# FAIRNESS-AWARE DOMAIN GENERALIZATION UNDER COVARIATE AND DEPENDENCE SHIFTS

## ABSTRACT

Achieving the generalization of an invariant classifier from source domains to shifted target domains while simultaneously considering model fairness is a substantial and complex challenge in machine learning. Existing domain generalization research typically attributes domain shifts to concept shift, which relates to alterations in class labels, and covariate shift, which pertains to variations in data styles. In this paper, by introducing another form of distribution shift, known as dependence shift, which involves variations in fair dependence patterns across domains, we propose a novel domain generalization approach that addresses domain shifts by considering both covariate and dependence shifts. We assert the existence of an underlying transformation model can transform data from one domain to another. By generating data in synthetic domains through the model, a fairness-aware invariant classifier is learned that enforces both model accuracy and fairness in unseen domains. Extensive empirical studies on four benchmark datasets demonstrate that our approach surpasses state-of-the-art methods. Anonymous link to the code for review purposes: https://anonymous.4open.science/r/FDDG-57DC.

## 1 INTRODUCTION

While modern fairness-aware machine learning techniques have demonstrated significant success in various applications (Zemel et al., 2013; Zhao et al., 2021; Wu et al., 2019), their primary objective is to facilitate equitable decision-making, ensuring fairness across all demographic groups, regardless of sensitive attributes, such as race and gender. Nevertheless, state-of-the-art methods can encounter severe shortcomings during the inference phase, mainly due to poor generalization when the spurious correlation deviates from the patterns seen in the training data. This correlation can manifest either between model outcomes and sensitive attributes (Creager et al., 2021; Oh et al., 2022) or between model outcomes and non-semantic data features (Pham et al., 2023). This issue originates from the existence of out-of-distribution (OOD) data, resulting in catastrophic failures.

Over the past decade, the machine learning community has made significant strides in studying the OOD generalization (or domain generalization, DG) problem and attributing the cause of the poor generalization to the distribution shifts from source domains to target domains. There are two dominant shift types (Moreno-Torres et al., 2012): concept shift and covariate shift. Concept shift refers to OOD samples drawn from a distribution with semantic change *e.g., dog v.s. cat*, and covariate shift characterizes the extent to which the distributions of data features differ across domains *e.g., photo v.s. cartoon*. While a variety of DG approaches have been explored, these methods often exhibit two specific limitations: (1) they predominantly address a single type of distribution shift (either concept or covariate shift) and disregard the significance of integrating fairness considerations into the learning process (Arjovsky et al., 2019; Chen et al., 2020; Krueger et al., 2021; Lu et al., 2021; Robey et al., 2021; Zhang et al., 2022), or (2) existing methods assume that the mentioned distribution shifts remain static, and there exist distinct fairness dependence between model outcomes and sensitive attributes in the source and target domains (Creager et al., 2021; Oh et al., 2022). Recently, *Pham et al.,* (Pham et al., 2023) address the DG problem with a focus on model fairness and accuracy in the presence of covariate shift. However, this work assumes that the fair dependence patterns across domains remain constant. Hence, there is a need for research that delves into the fairness-aware DG problem in the context of covariate shift and accounts for the

**Source Domain 1 (Photo)**  **Source Domain 2 (Cartoon)**  **Target Domain (Painting)**

**Fairness dependence between Dog/Cat and Grass/Couch: 0.25 (Photo); 0.5 (Cartoon); 0.25 (Painting)**

Figure 1: Illustration of fairness-aware domain generalization problems. Different data domains correspond to different image styles ("Photo", "Cartoon", and "Painting"). Each domain is associated with various fair dependencies between class labels ("Dog" and "Cat") and sensitive attributes ("Grass" and "Couch"), estimated using the demographic parity in Appendix C.3. In the Photo domain, (mostly) dogs in the grass and cats in couches. In the Painting domain, (mostly) dogs in couches and cats in the grass.

varying fairness dependence between source and target domains. Details and more related works are summarized in Appendix A.

In this paper, in addition to concept shift and covariate shift, inspired by (Creager et al., 2021), we first introduce a novel type of shift known as "dependence shift" for DG problems. The dependence shift is characterized by a disparity in the dependence between sensitive attributes and sample classes within source domains as compared to target domains. As illustrated in Figure 1, in the source domain 1, images labeled with "Dog" exhibit a strong correlation with the sensitive attribute "Grass", whereas "Cat" images are correlated with "Couch". However, in the target domain, this correlation between image labels and sensitive attributes is reversed and shifted from source domains. Furthermore, we define the fairness-aware DG problem within a broader context, where we account for two types of shifts, covariate shift and dependence shift, that occur when transferring from observed source domains to an unknown and inaccessible target domain. As shown in Fig. 1, various image styles ("Photo", "Cartoon" and "Painting") specify different data domains, and "Grass" and "Couch" correspond to sensitive attributes. The goal of the problem is to find a classifier that remains invariant in classifying between "Dog" and "Cat" across the observed source domains and subsequently enhance its generalization performance when faced with a target domain that is unseen during training characterized by different stylistic variations and fairness dependencies. To tackle the problem, we introduce a new framework, which we called *Fair Disentangled Domain Generalization* (FDDG). The key idea in our framework revolves around understanding transformations that account for both covariate and dependence shifts, enabling the mapping of data between domains, and then subsequently enforcing invariance by generating synthetic domains through these transformations. We leverage this framework to systematically define the fairness-aware DG problem as a semi-infinite constrained optimization problem. We then apply this re-formulation to demonstrate that a tight approximation of the problem can be achieved by solving the empirical, parameterized dual for this problem. Moreover, we develop a novel interpretable bound focusing on fairness within a target domain, considering the DG arising from both covariate and dependence shifts. Finally, extensive experimental results on the proposed new algorithm show that our algorithm significantly outperforms state-of-the-art baselines on several benchmarks.

**Contributions**. Our main contributions are summarized

- To our knowledge, we are the first to introduce a fairness-aware DG problem within a framework that accommodates inter-domain variations arising from two distinct types of distribution shifts, covariate shift and dependence shift.

- We reformulate the problem to a novel constrained learning problem. We further establish duality gap bounds for the empirically parameterized dual of this problem and develop a novel upper bound that specifically addresses fairness within a target domain while accounting for the domain generalization stemming from both covariate and dependence shifts.

- We present a novel algorithm designed to address the fairness-aware DG problem. This algorithm enforces invariance across unseen target domains by utilizing generative models derived from the observed source domains.

- Comprehensive experiments are conducted to verify the effectiveness of FDDG. We empirically show that our algorithm significantly outperforms state-of-the-art baselines on four benchmarks.

## 2   PRELIMINARIES

**Notations.** Let $\mathcal{X} \subseteq \mathbb{R}^d$ denote a feature space, $\mathcal{Z} = \{-1, 1\}$ is a sensitive label space, and $\mathcal{Y} = \{0, 1\}$ is a label space for classification. Let $\mathcal{C} \subseteq \mathbb{R}^c$, $\mathcal{A} \subseteq \mathbb{R}^a$, and $\mathcal{S} \subseteq \mathbb{R}^s$ be the latent content, sensitive and style spaces, respectively, induced from $\mathcal{X}$ and $\mathcal{A}$ by an underlying transformation model $T : \mathcal{X} \times \mathcal{Z} \times \mathcal{E} \to \mathcal{X} \times \mathcal{Z}$. We use $X, Z, Y, C, A, S$ to denote random variables that take values in $\mathcal{X}, \mathcal{Z}, \mathcal{Y}, \mathcal{C}, \mathcal{A}, \mathcal{S}$ and $\mathbf{x}, z, y, \mathbf{c}, \mathbf{a}, \mathbf{s}$ the realizations. A domain $e \in \mathcal{E}$ is specified by distribution $\mathbb{P}(X^e, Z^e, Y^e) : \mathcal{X} \times \mathcal{Z} \times \mathcal{Y} \to [0, 1]$. A classifier $f$ in a class space $\mathcal{F}$ denotes $f \in \mathcal{F} : \mathcal{X} \to \mathcal{Y}$.

**Fairness Notions.** When learning a fair classifier $f \in \mathcal{F}$ that focuses on statistical parity across different sensitive subgroups, the fairness criteria require the independence between the sensitive random variables $Z$ and the predicted model outcome $f(X)$ (Dwork et al., 2011). Addressing the issue of preventing group unfairness can be framed as the formulation of a constraint. This constraint mitigates bias by ensuring that $f(X)$ aligns with the ground truth $Y$, fostering equitable outcomes.

**Definition 1** (Group Fairness Notion (Wu et al., 2019; Lohaus et al., 2020)). *Given a dataset $\mathcal{D} = \{(\mathbf{x}_i, z_i, y_i)\}_{i=1}^{|\mathcal{D}|}$ sampled i.i.d. from $\mathbb{P}(X, Z, Y)$, a classifier $f \in \mathcal{F} : \mathcal{X} \to \mathcal{Y}$ is fair when the prediction $\hat{Y} = f(X)$ is independent of sensitive random variable $Z$. To get rid of the indicator function and relax the exact values, a linear approximated form of the difference between sensitive subgroups is defined as*

$$\rho(\hat{Y}, Z) = \left| \mathbb{E}_{\mathbb{P}(X,Z)} g(\hat{Y}, Z) \right| \quad where \quad g(\hat{Y}, Z) = \frac{1}{p_1(1-p_1)} \left( \frac{Z+1}{2} - p_1 \right) \hat{Y} \qquad (1)$$

*$p_1$ and $1 - p_1$ are the proportion of samples in the subgroup $Z = 1$ and $Z = -1$, respectively.*

Specifically, when $p_1 = \mathbb{P}(Z = 1)$ and $p_1 = \mathbb{P}(Z = 1, Y = 1)$, the fairness notion $\rho(\hat{Y}, Z)$ is defined as the difference of demographic parity (DP) and the difference of equalized odds (EO), respectively (Lohaus et al., 2020). In the paper, we will present the results under DP, while the framework can be generalized to multi-class, multi-sensitive attributes and other fairness notions. Strictly speaking, a classifier $f$ is fair over subgroups if it satisfies $\rho(\hat{Y}, Z) = 0$.

**Problem Setting.** We consider a set of data domains $\mathcal{E}$, where each domain $e \in \mathcal{E}$ corresponds to a distinct data subset $\mathcal{D}^e = \{(\mathbf{x}_i^e, z_i^e, y_i^e)\}_{i=1}^{|\mathcal{D}^e|}$ sampled i.i.d. from $\mathbb{P}(X^e, Z^e, Y^e)$. Given a dataset $\mathcal{D} = \{\mathcal{D}^e\}_{e \in \mathcal{E}}$, it is partitioned into multiple source domains $\mathcal{E}_s \subset \mathcal{E}$ and unknown target domains which are inaccessible during training. Therefore, given samples from finite source domains $\mathcal{E}_s$, the goal of fairness-aware domain generalization problems is to learn a classifier $f \in \mathcal{F}$ that is generalizable across all possible domains.

**Problem 1** (Fairness-aware Domain Generalization). *Let $\mathcal{E}_s \subset \mathcal{E}$ be a finite subset of source domains and assume that, for each $e \in \mathcal{E}_s$, we have access to its corresponding dataset $\mathcal{D}^e = \{(\mathbf{x}_i^e, z_i^e, y_i^e)\}_{i=1}^{|\mathcal{D}^e|}$ sampled i.i.d from $\mathbb{P}(X^e, Z^e, Y^e)$. Given a classifier set $\mathcal{F}$ and a loss function $\ell : \mathcal{Y} \times \mathcal{Y} \to \mathbb{R}$, the goal is to learn a fair classifier $f \in \mathcal{F}$ for any $\mathcal{D}^e \in \mathcal{D}_s = \{\mathcal{D}^e\}_{e \in \mathcal{E}_s}$ that minimizes the worst-case risk over all domains in $\mathcal{E}$ satisfying a group fairness constraint:*

$$\min_{f \in \mathcal{F}} \max_{e \in \mathcal{E}} \mathbb{E}_{\mathbb{P}(X^e, Z^e, Y^e)} \ell(f(X^e), Y^e), \quad subject \ to \ \rho(f(X^e), Z^e) = 0. \qquad (2)$$

The goal of Prob. 1 is to seek a fair classifier $f$ that generalizes from the given finite set of source domains $\mathcal{E}_s$ to give a good generalization performance on $\mathcal{E}$. Since we do not assume data from $\mathcal{E} \backslash \mathcal{E}_s$ is accessible, it makes this problem challenging to solve.

Another challenge is how closely the data distributions in unknown target domains match those in the observed source domains. In (Moreno-Torres et al., 2012), there are two forms of distribution shifts: concept shift, where the instance conditional distribution $\mathbb{P}(Y^e | X^e, Z^e)$ varies across different domains, and covariate shift, where the marginal distributions over instance $\mathbb{P}(X^e)$ are various. Yet, neither of these shifts captures the degree to which the distribution shifts with regard to model fairness. Therefore, we introduce a novel variation, dependence shift, where the dependence $\rho(Y^e, Z^e)$ between sensitive attributes and sample classes differs across domains.

**Definition 2** (Covariate Shift (Robey et al., 2021) and Dependence Shift). *In Prob. 1, covariate shift occurs when environmental variation is attributed to disparities in the marginal distributions across*

*instances $\{\mathbb{P}(X^e)\}_{e \in \mathcal{E}}$. On the other hand, Prob. 1 exhibits a dependence shift when environmental variation arises from alterations in the sensitive dependence $\rho(Y^e, Z^e)$.*

## 3 FAIRNESS-AWARE DISENTANGLED DOMAIN GENERALIZATION

DG tasks can generally be characterized by one form of the three distribution shifts. In this paper, we restrict the scope of our framework to focus on Prob. 1 in which inter-domain variation is due to covariate shift and dependence shift simultaneously through an underlying transformation model $T$.

**Assumption 1** (Transformation Model). *We assume that, $\forall e_i, e_j \in \mathcal{E}, e_i \neq e_j$, there exists a measurable function $T : \mathcal{X} \times \mathcal{Z} \times \mathcal{E} \rightarrow \mathcal{X} \times \mathcal{Z}$, referred as transformation model, that transforms instances from domain $e_i$ to $e_j$, denoted $(X^{e_j}, Z^{e_j}) = T(X^{e_i}, Z^{e_i}, e_j)$.*

Under Assump. 1, a data subset $\mathcal{D}^{e_j} \in \mathcal{D}$ of domain $e_j$ can be regarded as generated from another data subset $\mathcal{D}^{e_i}$ through the transformation model $T$ by altering $(X^{e_i}, Z^{e_i})$ to $(X^{e_j}, Z^{e_j})$. Building upon the insights from existing DG literature (Zhang et al., 2022; Zhao et al., 2023; Lin et al., 2023), we define $T$ with a specific emphasis on disentangling the variation in data features across domains into latent spaces with three factors. For specific information about the design of $T$ and the learning algorithm, please refer to Sec. 4.

**Assumption 2** (Multiple Latent Factors). *Given dataset $\mathcal{D}^e = \{(\mathbf{x}_i^e, z_i^e, y_i^e)\}_{i=1}^{|\mathcal{D}^e|}$ sampled i.i.d. from $\mathbb{P}(X^e, Z^e, Y^e)$ in domain $e \in \mathcal{E}$, we assume that each instance $\mathbf{x}_i^e$ is generated from*

- *a latent content factor $\mathbf{c} \in \mathcal{C}$, where $\mathcal{C} = \{\mathbf{c}_{y=0}, \mathbf{c}_{y=1}\}$ refers to a content space;*

- *a latent sensitive factor $\mathbf{a} \in \mathcal{A}$, where $\mathcal{A} = \{\mathbf{a}_{z=1}, \mathbf{a}_{z=-1}\}$ refers to a sensitive space;*

- *a latent style factor $\mathbf{s}^e$, where $\mathbf{s}^e$ is specific to the individual domain $e$.*

*We assume that the content and sensitive factors in $\mathcal{C}$ and $\mathcal{A}$ do not change across domains. Each domain $e$ over $\mathbb{P}(X^e, Z^e, Y^e)$ is represented by a style factor $\mathbf{s}^e$ and the dependence score $\rho^e = \rho(Y^e, Z^e)$, denoted $e := (\mathbf{s}^e, \rho^e)$, where $\mathbf{s}^e$ and $\rho^e$ are unique to the domain $e$.*

Note that Assump. 2 is similarly related to the one made in (Zhang et al., 2022; Robey et al., 2021; Huang et al., 2018; Liu et al., 2017). In our paper, with a focus on group fairness, we expand upon the assumptions of existing works by introducing three latent factors. Because we assume the instance conditional distribution $\mathbb{P}(Y|X, Z)$ remains consistent across domains (*i.e.,* there is no concept shift taking place), under Assumps. 1 and 2, if two instances $(\mathbf{x}^{e_i}, z^{e_i}, y)$ and $(\mathbf{x}^{e_j}, z^{e_j}, y)$ where $e_i, e_j \in \mathcal{E}, i \neq j$ share the same class label, then the latter instance can be reconstructed from the former using $T$. Specifically, $T$ constructs $\mathbf{x}^{e_j}$ using the content factor of $\mathbf{x}^{e_i}$, the sensitive factor of $\mathbf{x}^{e_j}$, and the style factor of $\mathbf{x}^{e_j}$. Additionally, $T$ constructs $z^{e_j}$ by employing the sensitive factor of $\mathbf{x}^{e_j}$. For fairness-aware invariant learning, we make the following assumption.

**Assumption 3** (Fairness-aware Domain Shift). *We assume that inter-domain variation is characterized by the covariate shift and dependence shift in $\mathbb{P}(X^e)$ and $\rho(Y^e, Z^e), \forall e \in \mathcal{E}$. As a consequence, we assume that $\mathbb{P}(Y^e|X^e, Z^e)$ is stable across domains. Given a domain transformation function $T$, for any $\mathbf{x} \in \mathcal{X}, z \in \mathcal{Z}$, and $y \in \mathcal{Y}$, it holds that*

$$\mathbb{P}(Y^{e_i} = y|X^{e_i} = \mathbf{x}^{e_i}, Z^{e_i} = z^{e_i}) = \mathbb{P}(Y^{e_j} = y|(X^{e_j}, Z^{e_j}) = T(\mathbf{x}^{e_i}, z^{e_i}, e_j)), \forall e_i, e_j \in \mathcal{E}, i \neq j$$

In Assump. 3, the domain shift captured by $T$ would characterize the mapping from the underlying distributions $\mathbb{P}(X^{e_i})$ and $\rho(Y^{e_i}, Z^{e_i})$ over $\mathcal{D}^{e_i}$ to the distribution $\mathbb{P}(X^{e_j})$ and $\rho(Y^{e_j}, Z^{e_j})$ of samples from a different data domain $\mathcal{D}^{e_j}$, respectively. With this in mind and under Assump. 3, we introduce a new definition of fairness-aware invariance with respect to the variation captured by $T$ and satisfying the group fair constraint introduced in Defn. 1.

**Definition 3** (Fairness-aware $T$-Invariance). *Given a transformation model $T$, a fairness-aware classifier $f \in \mathcal{F}$ is domain invariant if it holds*

$$f(\mathbf{x}^{e_i}) = f(\mathbf{x}^{e_j}), \quad and \quad \rho(f(X^{e_i}), Z^{e_i}) = \rho(f(X^{e_j}), Z^{e_j}) = 0 \tag{3}$$

*almost surely when $(\mathbf{x}^{e_j}, z^{e_j}) = T(\mathbf{x}^{e_i}, z^{e_i}, e_j)$, $\mathbf{x}^{e_i} \sim \mathbb{P}(X^{e_i})$, $\mathbf{x}^{e_j} \sim \mathbb{P}(X^{e_j})$, and $e_i, e_j \in \mathcal{E}$.*

Defn. 3 is crafted to enforce invariance on the predictions generated by $f$ directly. We expect a prediction to yield the same prediction for any realization of data under $T$ while being aware of group fairness.

**Problem 2** (Fairness-aware Disentanglement for Domain Generalization). *Under Defn. 3 and Assump. 3, if we restrict $\mathcal{F}$ of Prob. 1 to the set of invariant fairness-aware classifiers, the Prob. 1 is equivalent to the following problem*

$$P^\star \triangleq \min_{f \in \mathcal{F}} R(f) \triangleq \mathbb{E}_{\mathbb{P}(X^{e_i}, Z^{e_i}, Y^{e_i})} \ell(f(X^{e_i}), Y^{e_i}) \tag{4}$$

$$\textit{subject to } f(\mathbf{x}^{e_i}) = f(\mathbf{x}^{e_j}), \quad \rho(f(X^{e_i}), Z^{e_i}) = \rho(f(X^{e_j}), Z^{e_j}) = 0$$

*where $\mathbf{x}^{e_i} \sim \mathbb{P}(X^{e_i})$, $\mathbf{z}^{e_i} \sim \mathbb{P}(Z^{e_i})$, $(\mathbf{x}^{e_j}, z^{e_j}) = T(\mathbf{x}^{e_i}, z^{e_i}, e_j)$, $\forall e_i, e_j \in \mathcal{E}$, $i \neq j$.*

Similar to (Robey et al., 2021), Prob. 2 is not a composite optimization problem. Moreover, acquiring domain labels is often expensive or even unattainable, primarily due to privacy concerns. Consequently, under the assumptions of disentanglement-based invariance and domain shift, Problem 1 can be approximated to Problem 2 by removing the max operator. Furthermore, Prob. 2 offers a new and theoretically-principled perspective on Prob. 1, when data varies from domain to domain with respect to an underlying transformation model $T$. To optimize Prob. 2 is challenging because

- The strict equality constraints in Prob. 2 are difficult to enforce in practice;
- Enforcing constraints on deep networks is known to be a challenging problem due to non-convexity. Simply transforming them to regularization cannot guarantee satisfaction for constrained problems;
- As we have incomplete access to all domains $\mathcal{E}$ or $\mathbb{P}(X, Z, Y)$, it limits the ability to enforce fairness-aware $T$-invariance and further makes it hard to estimate $R(f)$.

Due to such challenges, we develop a tractable method for approximately solving Prob. 2 with optimality guarantees. To address the first challenge, we relax constraints in Prob. 2

$$P^\star(\gamma_1, \gamma_2) \triangleq \min_{f \in \mathcal{F}} R(f) \quad \text{subject to} \quad \delta^{e_i, e_j}(f) \leq \gamma_1, \epsilon^{e_i}(f) \leq \frac{\gamma_2}{2} \text{ and } \epsilon^{e_j}(f) \leq \frac{\gamma_2}{2} \tag{5}$$

where

$$\delta^{e_i, e_j}(f) \triangleq \mathbb{E}_{\mathbb{P}(X, Z)} d\big[f(X^{e_i}), f(X^{e_j} = T(X^{e_i}, Z^{e_i}, e_j))\big], \tag{6}$$

$$\epsilon^{e_i}(f) \triangleq \rho(f(X^{e_i}), Z^{e_i}), \quad \epsilon^{e_j}(f) \triangleq \rho(f(X^{e_j}), Z^{e_j}) \tag{7}$$

and $\forall e_i, e_j \in \mathcal{E}$, $i \neq j$. Here, $\gamma_1, \gamma_2 > 0$ are constants controlling the extent of relaxation and $d[\cdot]$ is a distance metric, *e.g.*, KL-divergence. When $\gamma_1 = \gamma_2 = 0$, Eqs. (4) and (5) are equivalent.

**Theorem 1** (Fairness Upper Bound of the Unseen Target Domain). *In accordance with Defn. 1 and Eq. (7), for any domain $e \in \mathcal{E}$, the fairness dependence under instance distribution $\mathbb{P}(X^e, Z^e, Y^e)$ with respect to the classifier $f \in \mathcal{F}$ is defined as:*

$$\epsilon^e(f) = \big|\mathbb{E}_{\mathbb{P}(X^e, Z^e)} g(f(X^e), Z^e)\big|$$

*With observed source domains $\mathcal{E}_s$, the dependence at any unseen target domain $e_t \in \mathcal{E}\backslash\mathcal{E}_s$ is upper bounded. $D[\cdot]$ is the Jensen-Shannon distance (Endres & Schindelin, 2003) metric.*

$$\epsilon^{e_t}(f) \leq \frac{1}{|\mathcal{E}_s|} \sum_{e_i \in \mathcal{E}_s} \epsilon^{e_i}(f) + \sqrt{2} \min_{e_i \in \mathcal{E}_s} D\big[\mathbb{P}(X^{e_t}, Z^{e_t}, Y^{e_t}), \mathbb{P}(X^{e_i}, Z^{e_i}, Y^{e_i})\big]$$

$$+ \sqrt{2} \max_{e_i, e_j \in \mathcal{E}_s} D\big[\mathbb{P}(X^{e_i}, Z^{e_i}, Y^{e_i}), \mathbb{P}(X^{e_j}, Z^{e_j}, Y^{e_j}\big]$$

*where $D[\mathbb{P}_1, \mathbb{P}_2] = \sqrt{\frac{1}{2}KL(\mathbb{P}_1||\frac{\mathbb{P}_1 + \mathbb{P}_2}{2}) + \frac{1}{2}KL(\mathbb{P}_2||\frac{\mathbb{P}_1 + \mathbb{P}_2}{2})}$ is JS divergence defined based on KL divergence.*

Since it is unrealistic to have access to the full distribution and we only have access to source domains, given data sampled from $\mathcal{E}_s$, we consider the empirical dual problem

$$D^\star_{\xi, N, \mathcal{E}_s}(\gamma_1, \gamma_2) \triangleq \max_{\lambda_1(e_i, e_j), \lambda_2(e_i, e_j)} \min_{\boldsymbol{\theta} \in \Theta} \hat{R}(\boldsymbol{\theta}) \tag{8}$$

$$+ \frac{1}{|\mathcal{E}_s|} \sum_{e_i, e_j \in \mathcal{E}_s} \Big[\lambda_1(e_i, e_j)\big(\hat{\delta}^{e_i, e_j}(\boldsymbol{\theta}) - \gamma_1\big) + \lambda_2(e_i, e_j)\big(\hat{\epsilon}^{e_i}(\boldsymbol{\theta}) + \hat{\epsilon}^{e_j}(\boldsymbol{\theta}) - \gamma_2\big)\Big]$$

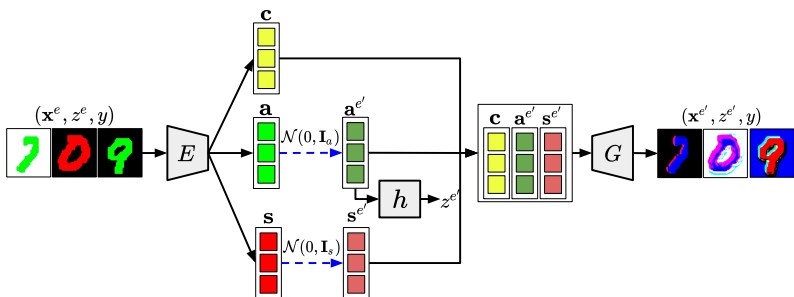

Figure 2: An overview of data generation in synthetic domains via an underlying transformation model $T$. Data are augmented through $T$ based on invariant content factors and randomly sampled sensitive and style factors that encode synthetic domains. We demonstrate the concept using the CCMNIST introduced in Sec. 5, where the domains are associated with various digit colors and the sensitive labels are determined by the background colors of each image.

where $\xi = \mathbb{E}_{\mathbb{P}(X)} ||f(\mathbf{x}) - \hat{f}(\mathbf{x}, \boldsymbol{\theta})||_\infty > 0$ is a constant bounding the difference between $f$ and its parameterized counterpart $\hat{f} : \mathcal{X} \times \Theta \to \mathbb{R}$ defined in the Definition 5.1 of (Robey et al., 2021). $\lambda_1(e_i, e_j), \lambda_2(e_i, e_j) > 0$ are dual variables. $\hat{R}(\boldsymbol{\theta}), \hat{\delta}^{e_i,e_j}(\boldsymbol{\theta}), \hat{\epsilon}^{e_i}(\boldsymbol{\theta})$ and $\hat{\epsilon}^{e_j}(\boldsymbol{\theta})$ are the empirical counterparts of $R(\hat{f}(\cdot, \boldsymbol{\theta})), \delta^{e_i,e_j}(\hat{f}(\cdot, \boldsymbol{\theta})), \epsilon^{e_i}(\hat{f}(\cdot, \boldsymbol{\theta}))$ and $\epsilon^{e_j}(\hat{f}(\cdot, \boldsymbol{\theta}))$, respectively. With such approximation on the dual problem in Eq. (8), the duality gap between $P^\star$ and $D^\star_{\xi,N,\mathcal{E}_s}(\gamma_1, \gamma_2)$ can be explicitly bounded.

**Theorem 2** (Fairness-aware Data-dependent Duality Gap). *Given $\xi > 0$, assuming $\{\hat{f}(\cdot, \boldsymbol{\theta}) : \boldsymbol{\theta} \in \Theta\} \subseteq \mathcal{F}$ has finite VC-dimension, with $M$ datapoints sampled from $\mathbb{P}(X, Z, Y)$ we have*

$$|P^\star - D^\star_{\xi,N,\mathcal{E}_s}(\boldsymbol{\gamma})| \le L||\boldsymbol{\gamma}||_1 + \xi k(1 + ||\boldsymbol{\lambda}^\star_p||_1) + O(\sqrt{\log(M)/M})$$

*where $\boldsymbol{\gamma} = [\gamma_1, \gamma_2]^T$; $L$ is the Lipschitz constant of $P^\star(\gamma_1, \gamma_2)$; $k$ is a small universal constant defined in Proposition 3 of Appendix E; and $\boldsymbol{\lambda}^\star_p$ is the optimal dual variable for a perturbed version of Eq.* (5).

The duality gap that arises when solving the empirical problem presented in Eq. (8) is minimal when the fairness-aware $T$-invariance in Defn. 3 margin $\boldsymbol{\gamma}$ is narrow, and the parametric space closely approximates $\mathcal{F}$. Proofs of Theorems 1 and 2 are provided in Appendix E.

## 4 AN EFFECTIVE ALGORITHM

**Fairness-aware DG via $T$.** Motivated by the theoretical insights in Sec. 3, we propose a simple but effective algorithm, namely FDDG. This algorithm consists of two stages. In the first stage, we train the transformation model $T$ using data from the source domains. In the second stage, we harness the power of $T$ to address the unconstrained dual optimization problem outlined in Eq. (8) through a series of primal-dual iterations.

Regarding the architecture of $T$, we expand upon the networks used in (Huang et al., 2018) by incorporating an additional output of the encoder $E : \mathcal{X} \times \mathcal{Z} \to \mathcal{C} \times \mathcal{A} \times \mathcal{S}$ for sensitive factors $\mathbf{a} \in \mathcal{A}$ and including a sensitive classifier $h : \mathcal{A} \to \mathcal{Z}$. A generator (decoder) $G : \mathcal{C} \times \mathcal{A} \times \mathcal{S} \to \mathcal{X} \times \mathcal{Z}$ is used to reconstruct instances from encoded latent factors. Following (Huang et al., 2018; Robey et al., 2021), the transformation model $T$ is trained using data in the observed source domains by applying reconstruction of them. Detailed training process of $T$ is provided in Appendix C.

Given the finite number of observed source domains, to enhance the generalization performance for unseen target domains, the invariant classifier $\hat{f}$ is trained by expanding the dataset with synthetic domains. These synthetic domains are created by introducing random instance styles and random fair dependencies within the domain. As described in Fig. 2, the sensitive factor $\mathbf{a}^{e'}$ and the style factor $\mathbf{s}^{e'}$ are randomly sampled from their prior distributions $\mathcal{N}(0, \mathbf{I}_a)$ and $\mathcal{N}(0, \mathbf{I}_s)$, respectively. Along with the unchanged content factor $\mathbf{c}$, they are further passed through $G$ to generate a new instance within a novel domain. Under Assump. 3 and Defn. 3, according to Eqs. (6) and (7), data

---

**Algorithm 1** Fair Disentangled Domain Generalization.

---

**Require**: Pretained encoder $E$, decoder $G$ and sensitive classifier $h$ within $T$.
**Initialize**: primal and dual learning rate $\eta_p, \eta_d$, empirical constant $\gamma_1, \gamma_2$.

---

1: **repeat**
2:   **for** minibatch $\mathcal{B} = \{(\mathbf{x}_i, z_i, y_i)\}_{i=1}^m \subset \mathcal{D}_s$ **do**
3:     $\mathcal{L}_{cls}(\boldsymbol{\theta}) = \frac{1}{m} \sum_{i=1}^m \ell(y_i, \hat{f}(\mathbf{x}_i, \boldsymbol{\theta}))$
4:     Initialize $\mathcal{L}_{inv}(\boldsymbol{\theta}) = 0$ and $\mathcal{B}' = [\,]$
5:     **for** each $(\mathbf{x}_i, z_i, y_i)$ in the minibatch **do**
6:       Generate $(\mathbf{x}_j, z_j, y_j) = \text{DATAAUG}(\mathbf{x}_i, z_i, y_i)$ and add it to $\mathcal{B}'$
7:       $\mathcal{L}_{inv}(\boldsymbol{\theta}) += \frac{1}{m} d[\hat{f}(\mathbf{x}_i, \boldsymbol{\theta}), \hat{f}(\mathbf{x}_j, \boldsymbol{\theta})]$
8:     **end for**
9:     $\mathcal{L}_{fair}(\boldsymbol{\theta}) = \left| \frac{1}{m} \sum_{(\mathbf{x}_i, z_i) \in \mathcal{B}} g(\hat{f}(\mathbf{x}_i, \boldsymbol{\theta}), z_i) \right| + \left| \frac{1}{m} \sum_{(\mathbf{x}_j, z_j) \in \mathcal{B}'} g(\hat{f}(\mathbf{x}_j, \boldsymbol{\theta}), z_j) \right|$
10:     $\mathcal{L}(\boldsymbol{\theta}) = \mathcal{L}_{cls}(\boldsymbol{\theta}) + \lambda_1 \cdot \mathcal{L}_{inv}(\boldsymbol{\theta}) + \lambda_2 \cdot \mathcal{L}_{fair}(\boldsymbol{\theta})$
11:     $\boldsymbol{\theta} \leftarrow \text{Adam}(\mathcal{L}(\boldsymbol{\theta}), \boldsymbol{\theta}, \eta_p)$
12:     $\lambda_1 \leftarrow \max\{[\lambda_1 + \eta_d \cdot (\mathcal{L}_{inv}(\boldsymbol{\theta}) - \gamma_1)], 0\}, \lambda_2 \leftarrow \max\{[\lambda_2 + \eta_d \cdot (\mathcal{L}_{fair}(\boldsymbol{\theta}) - \gamma_2)], 0\}$
13:   **end for**
14: **until** convergence
15: **procedure** DATAAUG$(\mathbf{x}, z, y)$
16:   $\mathbf{c}, \mathbf{a}, \mathbf{s} = E(\mathbf{x})$
17:   Sample $\mathbf{a}' \sim \mathcal{N}(0, I_a), \mathbf{s}' \sim \mathcal{N}(0, I_s)$
18:   $\mathbf{x}' = G(\mathbf{c}, \mathbf{a}', \mathbf{s}'), z' = h(\mathbf{a}')$
19:   **return** $(\mathbf{x}', z', y)$
20: **end procedure**

---

augmented in synthetic domains are required to maintain invariance in terms of accuracy and fairness with the data in the corresponding original domains.

**Walking Through Algorithm 1.** Our proposed implementation is shown in Algorithm 1 to solve the empirical dual Eq. (8). In lines 15-20, we describe the DATAAUG procedure that takes an example $(\mathbf{x}, z, y)$ as INPUT and returns an augmented example $(\mathbf{x}', z', y)$ from a new synthetic domain as OUTPUT. The augmented example has the same content factor as the input example but has different sensitive and style factors sampled from their associated prior distributions that encode a new synthetic domain. Lines 1-14 show the main training loop for FDDG. In line 6, for each example in the minibatch $\mathcal{B}$, we apply the procedure DATAAUG to generate an augmented example from a new synthetic domain described above. In line 7, we consider KL-divergence as the distance metric for $d[\cdot]$. All the augmented examples are stored in the set $\mathcal{B}'$. The Lagrangian dual loss function is defined based on $\mathcal{B}$ and $\mathcal{B}'$ in line 10. The primal parameters $\boldsymbol{\theta}$ and the dual parameters $\lambda_1$ and $\lambda_2$ are updated in lines 11-12.

## 5 EXPERIMENTS

**Settings.** We evaluate the performance of our FDDG on four benchmarks. To highlight each domain $e$ and its fair dependence score $\rho^e$, we summarize the statistic in Tab. 1. Three image datasets ccMNIST, FairFace (Karkkainen & Joo, 2021), YFCC100M-FDG (Thomee et al., 2016), and one tabular dataset New York Stop-and-Frisk (NYSF) (Koh et al., 2021) are conducted on FDDG against 17 state-of-the-art baseline methods that fall into two categories, state-of-the-art DG methods (RandAug[1], ERM (Vapnik, 1999), IRM (Arjovsky et al., 2019), GDRO (Sagawa et al., 2020), Mixup (Yan et al., 2020), MLDG (Li et al., 2018a), CORAL (Sun & Saenko, 2016), MMD (Li et al., 2018b), DANN (Ganin et al., 2016), CDANN (Li et al., 2018c), DDG (Zhang et al., 2022), and MBDG (Robey et al., 2021)) and fairness-aware methods in changing environments (DDG-FC, MBDG-FC[2], EIIL (Creager et al., 2021), FarconVAE (Oh et al., 2022), FATDM (Pham et al., 2023)). Three metrics are used for evaluation. Two of them are for fairness quantification, Demographic

---

[1]RandAug (or ColorJitter) is a naive built-in function in *Torch* used for image transformations. It randomly changes the brightness, contrast, saturation, and hue of given images.

[2]DDG-FC and MBDG-FC are two fairness-aware DG baselines that built upon DDG (Zhang et al., 2022) and MBDG (Robey et al., 2021), respectively. These extensions involve the straightforward addition of fairness constraints defined in Defn. 1 to the loss functions of the original models.

Table 1: Statistics summary of all datasets.

| Dataset | Domain, $e$ | Sensitive label, $Z$ | Class label, $Y$ | $(e, \rho^e), \forall e \in \mathcal{E}$ |
|---|---|---|---|---|
| ccMNIST | digit color | background color | digit label | (R, 0.11), (G, 0.43), (B, 0.87) |
| FairFace | race | gender | age | (B, 0.91), (E, 0.87), (I, 0.58), (L, 0.48), (M, 0.87), (S, 0.39), (W, 0.49) |
| YFCC100M-FDG | year | location | indoor/outdoor | $(d_0, 0.73), (d_1, 0.84), (d_2, 0.72)$ |
| NYSF | city | race | stop record | (R, 0.93), (B, 0.85), (M, 0.81), (Q, 0.98), (S, 0.88) |

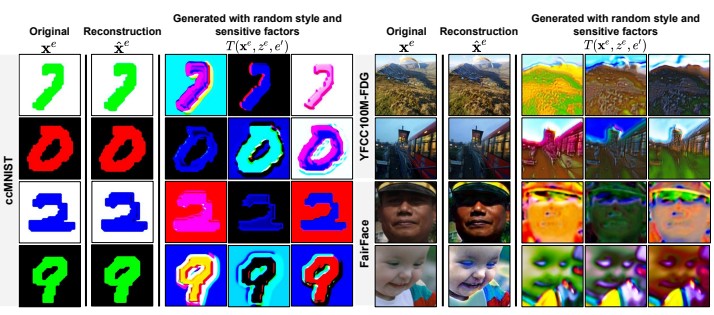

Figure 3: Visualizations for images under reconstruction and the transformation model $T$ with random style and sensitive factors.

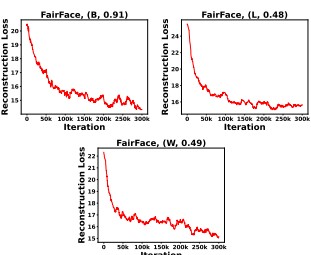

Figure 4: Tracking the change of reconstruction loss using the B/L/W domains of the FairFace dataset.

Parity (DP) (Dwork et al., 2011) and the Area Under the ROC Curve (AUC) between predictions of sensitive subgroups (Ling et al., 2003). Notice that the AUC metric is not the same as the one commonly used in classification based on TPR and FPR. The intuition behind this AUC is based on the nonparametric *Mann-Whitney U* test, in which a fair condition is defined as the probability of a randomly selected sample $\mathbf{x}_{-1}$ from one sensitive subgroup being greater than a randomly selected sample $\mathbf{x}_1$ from the other sensitive subgroup is equal to the probability of $\mathbf{x}_1$ being greater than $\mathbf{x}_{-1}$ (Zhao & Chen, 2019; Calders et al., 2013). A value of DP closer to 1 indicates fairness and 0.5 of AUC represents zero bias effect on predictions. Due to space limits, we defer a detailed description of the experimental settings (including datasets, baselines, evaluation metrics, *etc.*) in Appendix C and complete results on all baselines and datasets in Appendix F.

**Data Reconstruction and Generation via $T$.** To assess the effectiveness of the transformation model $T$, we visualize instance reconstruction and domain variation using image datasets in Fig. 3. The first column (Original) shows the images sampled from the datasets. In the second column (Reconstruction), we display images that are generated from latent factors encoded from the images in the first column. The images in the second column closely resemble those in the first column. We showcase the reconstruction loss using the FairFace dataset in Fig. 4. Images in the last three columns are generated using the content factors that were encoded from images in the first column. These images are generated with style and sensitive factors randomly sampled from their respective prior distributions. The images in the last three columns preserve the underlying semantic information of the corresponding samples in the first column. However, their style and sensitive attributes undergo significant changes. This demonstrates that the transformation model $T$ effectively extracts latent factors and produces diverse transformations of the provided data domains. More visualization and plot results are given in Appendix F.

**The Effectiveness of FDDG Across Domains in terms of Fairness and Accuracy.** Comprehensive experiments showcase that FDDG consistently outperforms baselines by a considerable margin. For all tables in the main paper and Appendix, results shown in each column represent performance on the target domain, using the rest as source domains. Due to space limit, selected results for three domains of FairFace are shown in Tab. 2, but the average results are based on all domains. Complete performance for all domains of datasets refers to Appendix F. As shown in Tab. 2, for the FairFace dataset, our method has the best accuracy and fairness level for the average DG performance over all the domains. More specifically, our method has better fairness metrics (4% for DP, 2% for AUC) and comparable accuracy (0.23% better) than the best of the baselines for individual metrics. As shown in Tab. 3, for YFCC100M-FDG, our method excels in fairness metrics (8% for DP, 5% for AUC) and comparable accuracy (0.35% better) compared to the best baselines.

**Ablation Studies.** We conduct three ablation studies to study the robustness of FDDG on FairFace. In-depth descriptions and the pseudocodes for these studies can be found in Appendix D. More results can be found in Appendix F. (1) In FDDG w/o sf, we modify the encoder

Table 2: Performance on for `FairFace`. **Bold** is the best and underline is the second best.

| Methods | DP ↑ / AUC ↓ / Accuracy ↑ | | | |
| --- | --- | --- | --- | --- |
| | (B, 0.91) | (L, 0.48) | (W, 0.49) | **Avg** |
| RandAug | 0.64±0.26 / 0.64±0.15 / 93.47±1.56 | 0.39±0.10 / 0.70±0.02 / 91.77±0.61 | 0.34±0.09 / 0.64±0.02 / 92.07±0.55 | 0.42 / 0.66 / 92.94 |
| ERM | 0.67±0.17 / 0.58±0.02 / 91.89±1.10 | 0.57±0.15 / 0.62±0.01 / 91.96±0.51 | 0.39±0.09 / 0.61±0.01 / 92.82±0.38 | 0.51 / 0.61 / 93.08 |
| IRM | 0.63±0.12 / 0.58±0.01 / 93.39±1.03 | 0.41±0.21 / 0.63±0.05 / 92.06±1.89 | 0.32±0.19 / 0.66±0.01 / 90.54±1.56 | 0.43 / 0.62 / 92.48 |
| GDRO | 0.71±0.16 / 0.57±0.02 / 89.81±1.10 | 0.54±0.15 / 0.62±0.01 / 91.59±0.51 | 0.48±0.09 / 0.60±0.01 / 92.50±0.38 | 0.55 / 0.60 / 92.55 |
| Mixup | 0.58±0.19 / 0.59±0.02 / 92.46±0.69 | 0.55±0.22 / 0.61±0.02 / 93.43±2.02 | 0.43±0.19 / 0.61±0.01 / 92.98±0.03 | 0.51 / 0.60 / 93.19 |
| DDG | 0.60±0.20 / 0.59±0.02 / 91.76±1.03 | 0.44±0.17 / 0.62±0.02 / 93.46±0.32 | 0.51±0.07 / 0.60±0.01 / 91.34±0.80 | 0.49 / 0.61 / 92.74 |
| MBDG | 0.60±0.15 / 0.58±0.01 / 91.29±1.41 | 0.56±0.09 / 0.61±0.01 / 93.49±0.97 | 0.30±0.04 / 0.62±0.01 / 91.05±0.53 | 0.50 / 0.60 / 92.71 |
| DDG-FC | 0.61±0.06 / 0.58±0.03 / 92.27±1.65 | 0.50±0.25 / 0.62±0.03 / 92.42±0.30 | 0.48±0.15 / 0.62±0.02 / 92.45±1.55 | 0.52 / 0.61 / 93.23 |
| MBDG-FC | 0.70±0.15 / 0.56±0.03 / 92.12±0.43 | 0.57±0.23 / 0.62±0.02 / 91.89±0.81 | 0.32±0.07 / 0.60±0.03 / 91.50±0.57 | 0.53 / 0.60 / 92.48 |
| EIIL | 0.88±0.07 / 0.59±0.05 / 84.75±2.16 | 0.49±0.07 / **0.59**±0.01 / 88.39±1.25 | 0.46±0.05 / 0.65±0.03 / 86.53±1.02 | 0.64 / 0.61 / 87.78 |
| FarconVAE | 0.93±0.03 / **0.54**±0.01 / 89.61±0.64 | **0.58**±0.05 / 0.60±0.05 / 88.70±0.71 | 0.51±0.07 / 0.60±0.01 / 86.40±0.42 | 0.66 / **0.58** / 88.46 |
| FATDM | 0.93±0.03 / 0.57±0.02 / 92.20±0.36 | 0.51±0.16 / 0.63±0.02 / 93.33±0.20 | 0.46±0.05 / 0.63±0.01 / 92.56±0.31 | 0.67 / 0.61 / 92.54 |
| FDDG | **0.94**±0.05 / 0.55±0.02 / **93.91**±0.33 | **0.58**±0.15 / **0.59**±0.01 / **93.73**±0.26 | **0.52**±0.17 / **0.58**±0.03 / **93.02**±0.50 | **0.70** / **0.58** / **93.42** |

Table 3: Performance on `YFCC100M-FDG`. (bold is the best; underline is the second best).

| Methods | DP ↑ / AUC ↓ / Accuracy ↑ | | | |
| --- | --- | --- | --- | --- |
| | ($d_0$, 0.73) | ($d_1$, 0.84) | ($d_2$, 0.72) | **Avg** |
| RandAug | 0.67±0.06 / 0.57±0.02 / 57.47±1.20 | 0.67±0.34 / 0.61±0.01 / 82.43±1.25 | 0.65±0.21 / 0.64±0.02 / 87.88±0.35 | 0.66 / 0.61 / 75.93 |
| ERM | 0.81±0.09 / 0.58±0.01 / 40.51±0.23 | 0.71±0.18 / 0.66±0.03 / 83.91±0.33 | 0.89±0.08 / 0.59±0.01 / 82.06±0.33 | 0.80 / 0.61 / 68.83 |
| IRM | 0.76±0.10 / 0.58±0.02 / 50.51±2.44 | 0.87±0.08 / 0.60±0.02 / 73.26±0.03 | 0.70±0.24 / 0.57±0.02 / 82.78±2.19 | 0.78 / 0.58 / 68.85 |
| GDRO | 0.80±0.05 / 0.59±0.01 / 53.43±2.29 | 0.73±0.22 / 0.60±0.01 / 87.56±2.20 | 0.79±0.13 / 0.65±0.02 / 83.10±0.64 | 0.78 / 0.62 / 74.70 |
| Mixup | 0.82±0.07 / 0.57±0.03 / 61.15±0.28 | 0.79±0.14 / 0.63±0.03 / 78.63±0.97 | 0.89±0.05 / 0.60±0.01 / 85.18±0.80 | 0.84 / 0.60 / 74.99 |
| DDG | 0.81±0.14 / 0.57±0.03 / 60.08±1.08 | 0.74±0.12 / 0.66±0.03 / 92.53±0.91 | 0.71±0.21 / 0.59±0.03 / 84.53±1.92 | 0.75 / 0.61 / 82.54 |
| MBDG | 0.79±0.15 / 0.58±0.01 / 60.46±1.90 | 0.73±0.07 / 0.67±0.01 / 94.36±0.23 | 0.71±0.11 / 0.59±0.03 / 93.48±0.65 | 0.74 / 0.61 / 82.77 |
| DDG-FC | 0.76±0.06 / 0.58±0.03 / 59.96±2.36 | 0.83±0.06 / 0.58±0.01 / **96.80**±1.28 | 0.82±0.09 / 0.59±0.01 / 86.38±2.45 | 0.80 / 0.58 / 81.04 |
| MBDG-FC | 0.80±0.13 / 0.58±0.01 / 62.31±0.13 | 0.72±0.09 / 0.63±0.01 / 94.73±2.09 | 0.80±0.07 / **0.53**±0.01 / 87.78±2.11 | 0.77 / 0.58 / 81.61 |
| EIIL | **0.87**±0.11 / 0.55±0.02 / 56.74±0.60 | 0.76±0.05 / 0.54±0.03 / 68.99±0.91 | 0.87±0.06 / 0.78±0.03 / 72.19±0.75 | 0.83 / 0.62 / 65.98 |
| FarconVAE | 0.67±0.06 / 0.61±0.03 / 51.21±0.61 | 0.90±0.06 / 0.59±0.01 / 72.40±2.13 | 0.85±0.12 / 0.55±0.01 / 74.20±2.46 | 0.81 / 0.58 / 65.93 |
| FATDM | 0.80±0.10 / 0.55±0.01 / 61.56±0.89 | 0.88±0.08 / 0.56±0.01 / 90.00±0.66 | 0.86±0.10 / 0.60±0.02 / 89.12±1.30 | 0.84 / 0.57 / 80.22 |
| FDDG | **0.87**±0.09 / **0.53**±0.01 / **62.56**±2.25 | **0.94**±0.05 / **0.52**±0.01 / 93.36±1.70 | **0.93**±0.03 / **0.53**±0.02 / 93.43±0.73 | **0.92** / **0.53** / **83.12** |

Table 4: Ablation studies results on `FairFace`.

| Methods | DP ↑ / AUC ↓ / Accuracy ↑ | | | |
| --- | --- | --- | --- | --- |
| | (B, 0.91) | (L, 0.48) | (W, 0.49) | **Avg** |
| FDDG w/o sf | 0.68±0.18 / 0.57±0.02 / 93.07±0.68 | 0.47±0.07 / 0.63±0.01 / 92.62±0.93 | 0.35±0.26 / 0.58±0.01 / 92.18±0.46 | 0.49 / 0.59 / 93.08 |
| FDDG w/o $T$ | 0.83±0.08 / 0.56±0.01 / 92.81±0.81 | 0.53±0.03 / **0.59**±0.01 / 91.19±0.57 | 0.52±0.23 / 0.59±0.01 / 90.78±0.31 | 0.58 / 0.59 / 91.65 |
| FDDG w/o fc | 0.59±0.16 / 0.58±0.01 / 92.92±1.35 | 0.40±0.07 / 0.70±0.02 / 92.96±0.85 | 0.34±0.08 / 0.72±0.03 / 91.88±0.67 | 0.42 / 0.66 / 93.01 |

within $T$ by restricting its output to only latent content and style factors. (2) FDDG w/o $T$ skips data augmentation in synthetic domains via $T$ and results are conducted only based $f$ constrained by fair notions outlined in Defn. 1. (3) In FDDG w/o fc, the fair constraint on $f$ is not included, and we eliminate the $\mathcal{L}_{fair}$ in line 9 of Algorithm 1. We include the performance of such ablation studies in Tab. 4. The results illustrate that when data is disentangled into three factors and the model is designed accordingly, it can enhance DG performance due to covariate and dependence shifts. Generating data in synthetic domains with randomly fair dependence patterns proves to be an effective approach for ensuring fairness invariance across domains.

## 6 CONCLUSION

Differing from established domain generalization research, which attributes distribution shifts from source domains to target domains to concept shift and covariate shift, we introduce a novel form of distribution shift known as dependence shift. This shift pertains to varying dependence patterns between model outcomes and sensitive attributes in different domains. Furthermore, we introduce a novel approach to fairness-aware learning designed to tackle the challenges of domain generalization when confronted with both covariate shift and dependence shift simultaneously. In our pursuit of learning a fairness-aware invariant classifier across domains, we assert the existence of an underlying transformation model that can transform instances from one domain to another. This model plays a crucial role in achieving fairness-aware domain generalization by generating instances in synthetic domains characterized by novel data styles and fair dependence patterns. We present a practical and tractable algorithm, accompanied by comprehensive theoretical analyses and exhaustive empirical studies. We showcase the algorithm's effectiveness through rigorous comparisons with state-of-the-art baselines.

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
