# Supplementary Materials

## A   RELATED WORKS

**Domain generalization.** Addressing the challenge of domain shift and the absence of OOD data has led to the introduction of several state-of-the-art methods in the domain generalization field (Vapnik, 1999; Arjovsky et al., 2019; Zhang et al., 2022; Robey et al., 2021). These methods are designed to enable deep learning models to possess intrinsic generalizability, allowing them to adapt effectively from one or multiple source domains to target domains characterized by unknown distributions (Volpi et al., 2021). They encompass various techniques, such as aligning source domain distributions to facilitate domain-invariant representation learning (Li et al., 2018b), subjecting the model to domain shift during training through meta-learning (Li et al., 2018a), and augmenting data with domain analysis, among others (Zhou et al., 2020), and so on. In the context of the number of source domains, a significant portion of research (Zhang et al., 2022; Robey et al., 2021; Blanchard et al., 2011) has focused on the multi-source setting. This setting assumes the availability of multiple distinct but relevant domains for the generalization task. As mentioned in (Blanchard et al., 2011), the primary motivation for studying domain generalization is to harness data from multiple sources in order to unveil stable patterns. This entails learning representations that are invariant to the marginal distributions of data features, all while lacking access to the target data. Nevertheless, existing domain generalization methods tend to overlook the aspect of learning with fairness, where group fairness dependence patterns may not undergo changes across domains.

**Fairness learning for changing environments.** Two primary research directions aim to tackle fairness-aware machine learning in dynamic or changing environments. The first approach involves equality-aware monitoring methods (Kirton, 2019; Alonso et al., 2021; Pham et al., 2023; Rezaei et al., 2021; Singh et al., 2021; Giguere et al., 2022; Chen et al., 2022), which strive to identify and mitigate unfairness in a model's behavior by continuously monitoring its predictions. These methods adapt the model's parameters or structure when unfairness is detected. However, a significant limitation of such approaches is their assumption of invariant fairness levels across domains, which may not hold in real-world applications. The second approach (Oh et al., 2022; Creager et al., 2021) focuses on evaluating a model's fairness in a dynamic environment by treating shifted fairness levels as domain labels. However, it does not take into account distribution shifts in non-sensitive features.

In response to these limitations, this paper adopts a novel approach by attributing the distribution shift from source to target domains to both covariate shift and fairness dependence shift simultaneously. We aim to train a fairness-aware invariant classifier that can generalize effectively across domains, ensuring robust performance in terms of both model accuracy and maintaining fair dependence between predicted outcomes and sensitive attributes even under these shifts.

## B   NOTATIONS

For clear interpretation, we list the notations used in this paper and their corresponding explanation, as shown in Tab. 5.

## C   EXPERIMENTAL SETTINGS

### C.1   DATASETS.

We consider four datasets: `ccMNIST`, `FairFace`, `YFCC100M-FDG`, and `New York Stop-and-Frisk (NYSF)` to evaluate our FDDG against state-of-the-art baseline methods, where `NYSF` is a tabular data and the other three are image datasets.

**(a)** `ccMNIST` is a domain generalization benchmark created by colorizing digits and the backgrounds of the `MNIST` dataset (LeCun et al., 1998). `ccMNIST` consists of images of handwritten digits from 0 to 9. Similar to `ColoredMNIST` (Arjovsky et al., 2019), for binary classification, digits are labeled with 0 and 1 for digits from 0-4 and 5-9, respectively. `ccMNIST` contains three data domains, each characterized by a different digit color (*i.e.,* red, green, blue) with 70,000 images. Each image has a black or white background color as the sensitive label. The domains are

Table 5: Important notations and corresponding descriptions.

| Notations | Descriptions |
|---|---|
| $\mathcal{X}$ | input feature space |
| $\mathcal{Z}$ | sensitive space |
| $\mathcal{Y}$ | output space |
| $\mathcal{C}$ | parameterized latent space for content factors |
| $\mathcal{S}$ | parameterized latent space for style factors |
| $\mathcal{A}$ | parameterized latent space for sensitive factors |
| $\mathbf{c}$ | content factor |
| $\mathbf{s}$ | style factor |
| $\mathbf{a}$ | sensitive factor |
| $d[\cdot]$ | distance metric on output space |
| $\mathcal{D}$ | data batch |
| $\mathbf{x}$ | data features |
| $y$ | class label |
| $z$ | sensitive label |
| $f$ | classifier |
| $\mathcal{F}$ | model space of classifier |
| $\hat{f}$ | $\xi$-parameterization of $\mathcal{F}$ |
| $\hat{y}$ | predicted class label |
| $\Theta$ | parameter space |
| $g(Y, Z)$ | fairness metric on random variables $Y$ and $Z$ |
| $|\cdot|$ | absolute function |
| $p_1$ | empirical estimate of the proportion of samples in the group $z = 1$ |
| $e$ | data domain |
| $\mathcal{E}$ | set of data domains |
| $\mathcal{B}$ | sampled data batch |
| $T$ | domain transformation model |
| $E$ | encoder network |
| $G$ | decoder network |
| $\mathcal{L}$ | loss function |
| $\delta, \epsilon$ | expectation of the relaxed constraint |
| $h$ | sensitive label classifier |
| $\hat{z}$ | sensitive label predicted by $h$ |
| $\eta_p, \eta_d$ | primal and dual learning rate |
| $\lambda$ | dual variable |
| $\gamma$ | empirical constant |

constructed so that each domain has a different correlation between the class label and sensitive attribute (digit background colors), specifically 0.9 for the red domain, 0.7 for the green domain, and 0 for the blue domain.

**(b)** FairFace (Karkkainen & Joo, 2021) is a dataset that contains a balanced representation of different racial groups. It includes 108,501 images from seven racial categories: Black (B), East Asian (E), Indian (I), Latino (L), Middle Eastern (M), Southeast Asian (S), and White (W). In our experiments, we set each racial group as a domain, gender as the sensitive label, and age ($\geq$ or $<$ 50) as the class label.

**(c)** YFCC100M-FDG is an image dataset created by *Yahoo Labs* and released to the public in 2014. It is randomly selected from the YFCC100M (Thomee et al., 2016) dataset with a total of 90,000 images. For domain variations, YFCC100M-FDG is divided into three domains. Each contains 30,000 images from different year ranges, before 1999 ($d_0$), 2000 to 2009 ($d_1$), and 2010 to 2014 ($d_2$). The outdoor or Indoor tag is used as the binary class label for each image. Latitude and longitude coordinates, representing where images were taken, are translated into different continents. The continent North-America or non-North-America is used as the sensitive label (related to spatial disparity).

**(d)** `NYSF` (Koh et al., 2021) is a real-world dataset on policing in New York City in 2011. It documents whether a pedestrian who was stopped on suspicion of weapon possession would in fact possess a weapon. `NYSF` consists of records collected in five different sub-cities, Manhattan (M), Brooklyn (B), Queens (Q), Bronx (R), and Staten (S). We use cities as different domains. This data had a pronounced racial bias against African Americans, so we consider race (black or non-black) as the sensitive attribute.

## C.2 Baselines.

We compare the performance of our FDDG with 17 baseline methods that fall into three main categories:

- 12 state-of-the-art *domain generalizations* methods (RandAug, ERM (Vapnik, 1999), IRM (Arjovsky et al., 2019), GDRO (Sagawa et al., 2020), Mixup (Yan et al., 2020), MLDG (Li et al., 2018a), CORAL (Sun & Saenko, 2016), MMD (Li et al., 2018b), DANN (Ganin et al., 2016), CDANN (Li et al., 2018c), DDG (Zhang et al., 2022), and MBDG (Robey et al., 2021));
- 3 state-of-the-art *fairness-aware learning* methods in changing environments (EIIL (Creager et al., 2021), FarconVAE (Oh et al., 2022), and FATDM (Pham et al., 2023));
- 2 *naive fairness-aware variants* of DDG and MBDG, named DDG-FC and MBDG-FC, respectively, by simply adding fairness constraints in Defn. 1 to their classifiers.

Notice that the settings of EIIL and FarconVAE are different from this paper. Both methods characterize domain shift by a different level of correlation between the class label and sensitive features but completely ignore the variation in data features.

## C.3 Evaluation Metrics.

Three metrics are used for evaluation, and two of them are for fairness quantification.

- *Demographic Parity* (DP) (Dwork et al., 2011) is formalized as

  $$\text{DP} = k, \text{ if DP} \le 1; \text{DP} = 1/k, \text{ otherwise}, \quad \text{where } k = \mathbb{P}(\hat{Y} = 1 | Z = -1)/\mathbb{P}(\hat{Y} = 1 | Z = 1)$$

  This is also known as a lack of disparate impact (Feldman et al., 2015). A value closer to 1 indicates fairness.

- *The Area Under the ROC Curve* (AUC) (Calders et al., 2013) varies from zero to one, and it is symmetric around 0.5, which represents random predictability or zero bias effect on predictions.

  $$\text{AUC} = \frac{\sum_{(\mathbf{x}_i, z=-1, y_i) \in \mathcal{D}_{-1}} \sum_{(\mathbf{x}_j, z=1, y_j) \in \mathcal{D}_1} I\big(\mathbb{P}(\hat{y}_i = 1) > \mathbb{P}(\hat{y}_j = 1)\big)}{|\mathcal{D}_{-1}| \times |\mathcal{D}_1|}$$

  where $|\mathcal{D}_{-1}|$ and $|\mathcal{D}_1|$ represent sample size of subgroups $z = -1$ and $z = 1$, respectively. $I(\cdot)$ is the indicator function that returns 1 when its argument is true and 0 otherwise.

## C.4 Learning the Transformation Model

One goal of the transformation model $T$ is to disentangle an input instance from source domains into three factors in latent spaces by learning a set of encoders $E = \{E^m, E^s, E^c, E^a\}$ and decoders $G = \{G^i, G^o\}$ parameterized by $\{\boldsymbol{\theta}_m, \boldsymbol{\theta}_s, \boldsymbol{\theta}_c, \boldsymbol{\theta}_a\} \in \Theta$ and $\{\boldsymbol{\phi}_i, \boldsymbol{\phi}_o\} \in \Phi$, respectively. As shown in Fig. 5, the learning process of $T$ consists of two levels, an outer level and an inner level, where each level is associated with an auto-encoder system factorizing its corresponding input into two factors within two separated latent spaces. Specifically, in the outer level, an instance is first encoded to a semantic factor $\mathbf{m} \in \mathcal{M}$ and a style factor $\mathbf{s} \in \mathcal{S}$ through the corresponding encoders $E^m : \mathcal{X} \times \Theta \to \mathcal{M}$ and $E^s : \mathcal{X} \times \Theta \to \mathcal{S}$, respectively. In the inner level, the semantic factor $\mathbf{m}$ is further encoded to a content factor $\mathbf{c} \in \mathcal{C}$ and a sensitive factor $\mathbf{a} \in \mathcal{A}$, through encoders $E^c : \mathcal{M} \times \Theta \to \mathcal{C}$ and $E^a : \mathcal{M} \times \Theta \to \mathcal{A}$. For data reconstruction, two decoders $G^i : \mathcal{C} \times \mathcal{A} \times \Phi \to \mathcal{M}$ and $G^o : \mathcal{M} \times \mathcal{S} \times \Phi \to \mathcal{X}$ are introduced in the inner and outer levels. Inspired by image-to-image translation in computer vision (Huang et al., 2018; Liu et al., 2017), our total loss function

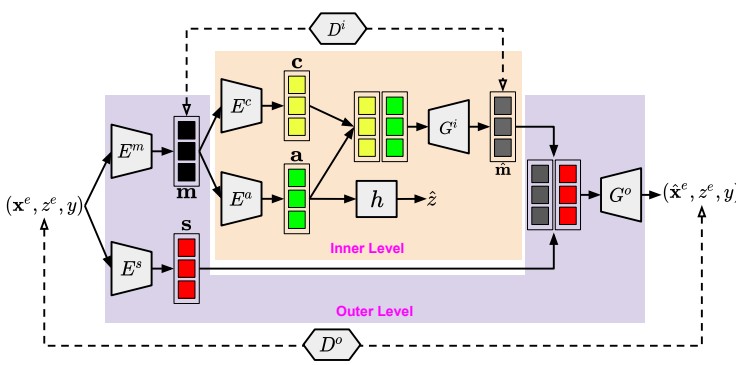

Figure 5: A two-level approach for leaning the transformation model $T$.

of learning such encoders and decoders comprises three components: a bidirectional reconstruction loss, a sensitive label prediction loss, and an adversarial loss.

**Reconstruction Loss** encourages learning reconstruction in two directions: (1) data→latent→data for *data reconstruction*, and (2) latent→data→latent for *factor reconstruction*. For simplicity, we omit parameters for encoders and decoders in the following equations. As for data reconstruction, in terms of the outer and inner levels, an instance $\mathbf{x}$ and its semantic factor $\mathbf{m}$ are required to be reconstructed, respectively.

$$\mathcal{L}_{recon}^{data} = \underbrace{\mathbb{E}_{\mathbf{x} \sim p(\mathbf{x})}\left[\left\|G^o\big(\hat{\mathbf{m}}, E^s(\mathbf{x})\big) - \mathbf{x}\right\|_1\right]}_{\text{data reconstruction}} + \underbrace{\mathbb{E}_{\mathbf{m} \sim p(\mathbf{m})}\left[\left\|G^i\big(E^c(\mathbf{m}), E^a(\mathbf{m})\big) - \mathbf{m}\right\|_1\right]}_{\text{reconstruction of semantic factors } \mathbf{m} \text{ (inner level)}}$$

where $\hat{\mathbf{m}} = G^i(\mathbf{c}, \mathbf{a}) = G^i\big(E^c(E^m(\mathbf{x})), E^a(E^m(\mathbf{x}))\big)$; $p(\mathbf{m})$ is given by $\mathbf{m} = E^m(\mathbf{x})$ and $\mathbf{x} \sim p(\mathbf{x})$. For factor reconstruction, $\mathbf{m}, \mathbf{s}, \mathbf{c}$ and $\mathbf{a}$ are encouraged to be reconstructed through some latent factors randomly sampled from the prior distributions.

$$\mathcal{L}_{recon}^{factor} = \underbrace{\mathbb{E}_{\mathbf{c} \sim p(\mathbf{c}), \mathbf{a} \sim \mathcal{N}(0, \mathbf{I}_a)}\left[\left\|E^c\big(G^i(\mathbf{c}, \mathbf{a})\big) - \mathbf{c}\right\|_1\right]}_{\text{reconstruction of content factors } \mathbf{c}} + \underbrace{\mathbb{E}_{\mathbf{c} \sim p(\mathbf{c}), \mathbf{a} \sim \mathcal{N}(0, \mathbf{I}_a)}\left[\left\|E^a\big(G^i(\mathbf{c}, \mathbf{a})\big) - \mathbf{a}\right\|_1\right]}_{\text{reconstruction of sensitive factors } \mathbf{a}}$$

$$+ \underbrace{\mathbb{E}_{\mathbf{m} \sim p(\mathbf{m}), \mathbf{s} \sim \mathcal{N}(0, \mathbf{I}_s)}\left[\left\|E^s\big(G^o(\mathbf{m}, \mathbf{s})\big) - \mathbf{s}\right\|_1\right] + \mathbb{E}_{\mathbf{c} \sim p(\mathbf{c}), \mathbf{s} \sim \mathcal{N}(0, \mathbf{I}_s), \mathbf{a} \sim \mathcal{N}(0, \mathbf{I}_a)}\left[\left\|E^s\big(G^o(G^i(\mathbf{c}, \mathbf{a}), \mathbf{s})\big) - \mathbf{s}\right\|_1\right]}_{\text{reconstruction of style factors } \mathbf{s}}$$

$$+ \underbrace{\mathbb{E}_{\mathbf{m} \sim p(\mathbf{m}), \mathbf{s} \sim \mathcal{N}(0, \mathbf{I}_s)}\left[\left\|E^m\big(G^o(\mathbf{m}, \mathbf{s})\big) - \mathbf{m}\right\|_1\right]}_{\text{reconstruction of semantic factors } \mathbf{m} \text{ (outer level)}}$$

where $p(\mathbf{c})$ is given by $\mathbf{c} = E^c(E^m(\mathbf{x}))$, $\mathbf{a} = E^a(E^m(\mathbf{x}))$, and $\mathbf{s} = E^s(\mathbf{x})$.

**Sensitive Loss.** Since a sensitive factor is causally dependent on the sensitive features of a datapoint, as shown in the inner level of Fig. 2, a simple classifier $h : \mathcal{A} \times \Theta \to \mathcal{Z}$ is trained and further it is used to predict the sensitive label using $\mathbf{a}$ in the second stage.

$$\mathcal{L}_{sens} = CrossEntropy(\mathbf{z}, \hat{\mathbf{z}}) \quad \text{where} \quad \hat{z} = h(\mathbf{a}, \boldsymbol{\theta}_z) = h(E^a(E^m(\mathbf{x})), \boldsymbol{\theta}_z)$$

**Adversarial Loss.** Motivated by the observation that GANs (Goodfellow et al., 2020) can improve data quality for evaluating the disentanglement effect in the latent spaces, we use GANs to match the distribution of reconstructed data to the same distribution. Followed by (Huang et al., 2018), data and semantic factors generated through encoders and decoders should be indistinguishable from the given ones in the same domain.

$$\mathcal{L}_{adv} = \underbrace{\mathbb{E}_{\mathbf{c} \sim p(\mathbf{c}), \mathbf{s} \sim \mathcal{N}(0, \mathbf{I}_s), \mathbf{a} \sim \mathcal{N}(\mathbf{0}, \mathbf{I}_a)}\left[\log\big(1 - D^o(G^o(\hat{\mathbf{m}}, \mathbf{s}))\big)\right] + \mathbb{E}_{\mathbf{x} \sim p(\mathbf{x})}\left[\log D^o(\mathbf{x})\right]}_{\text{outer level}}$$

$$+ \underbrace{\mathbb{E}_{\mathbf{c} \sim p(\mathbf{c}), \mathbf{a} \sim \mathcal{N}(\mathbf{0}, \mathbf{I}_a)}\left[\log\big(1 - D^i(G^i(\mathbf{c}, \mathbf{a}))\big)\right] + \mathbb{E}_{\mathbf{m} \sim p(\mathbf{m})}\left[\log D^i(\mathbf{m})\right]}_{\text{inner level}}$$

where $D^o : \mathcal{X} \times \Psi \to \mathbb{R}$ and $D^i : \mathcal{M} \times \Psi \to \mathbb{R}$ are the discriminators for the outer and inner levels parameterized by $\boldsymbol{\psi}_o \in \Psi$ and $\boldsymbol{\psi}_i \in \Psi$, respectively.

---

**Algorithm 2** Learning the Transformation Model $T$.

---

**Require**: learning rate $\alpha_1, \alpha_2, \alpha_3$, initial coefficients $\beta_d, \beta_f, \beta_z, \beta_g$.
**Initialize**: Parameter of encoders $\{\boldsymbol{\theta}_m, \boldsymbol{\theta}_s, \boldsymbol{\theta}_c, \boldsymbol{\theta}_a\}$, decoders $\{\boldsymbol{\phi}_i, \boldsymbol{\phi}_o\}$, sensitive classifier $\boldsymbol{\theta}_z$, and discriminators $\{\boldsymbol{\psi}_i, \boldsymbol{\psi}_o\}$.

1: **repeat**
2:      **for** minibatch $\{(\mathbf{x}_i, y_i, z_i)\}_{i=1}^q \in \mathcal{D}_s$ **do**
3:          Compute $\mathcal{L}_{total}$ for Stage 1 using Eq. (9).
4:          $\boldsymbol{\psi}_o, \boldsymbol{\psi}_i \leftarrow \text{Adam}(\beta_g \mathcal{L}_{adv}, \boldsymbol{\psi}_o, \boldsymbol{\psi}_i, \alpha_1)$
5:          $\boldsymbol{\theta}_m, \boldsymbol{\theta}_c, \boldsymbol{\theta}_s, \boldsymbol{\theta}_a, \boldsymbol{\phi}_o, \boldsymbol{\phi}_i \leftarrow \text{Adam}\big(\beta_d \mathcal{L}_{recon}^{data} + \beta_f \mathcal{L}_{recon}^{factor}, \boldsymbol{\theta}_m, \boldsymbol{\theta}_c, \boldsymbol{\theta}_s, \boldsymbol{\theta}_a, \boldsymbol{\phi}_o, \boldsymbol{\phi}_i, \alpha_2\big)$
6:          $\boldsymbol{\theta}_z \leftarrow \text{Adam}(\beta_z \mathcal{L}_{sens}, \boldsymbol{\theta}_z, \alpha_3)$
7:      **end for**
8: **until** convergence
9: **Return** $\{\boldsymbol{\theta}_m, \boldsymbol{\theta}_s, \boldsymbol{\theta}_c, \boldsymbol{\theta}_a, \boldsymbol{\theta}_z, \boldsymbol{\phi}_i, \boldsymbol{\phi}_o\}$

---

**Total Loss.** We jointly train the encoders, decoders, and discriminators to optimize the final objective, a weighted sum of the three loss terms.

$$\min_{E^m, E^s, E^c, E^a, G^i, G^o} \max_{D^i, D^o} \mathcal{L}_{total} = \beta_d \mathcal{L}_{recon}^{data} + \beta_f \mathcal{L}_{recon}^{factor} + \beta_z \mathcal{L}_{sens} + \beta_g \mathcal{L}_{adv} \tag{9}$$

where $\beta_d, \beta_f, \beta_z, \beta_g > 0$ are hyperparameters that control the importance of each loss term. To optimize, the learning algorithm is given in Algorithm 2.

## C.5 ARCHITECTURE DETAILS

We have two sets of networks. One is for ccMNIST, FairFace, and YFCC100M-FDG, and the other one is for the NYSF dataset.

For ccMNIST, FairFace, and YFCC100M-FDG datasets: All the images are resized to $224 \times 224$. $E^m$ and $E^c$'s structures are the same. Each of them is made of four convolution layers. The first one has 64 filters, and each of the others has 128 filters. The kernel sizes are $(7, 7), (4, 4), (3, 3), (3, 3)$ for layers 1 to 4, respectively. The stride of the second layer is $(2, 2)$, and the stride of all the other layers is $(1, 1)$. The activation function of the first three layers is ReLU. The last convolution layer does not have an activation function. $E^s$ and $E^a$'s structures are the same. Each of them is made of 6 convolution layers, and there is an adaptive average pooling layer with output size 1 between the last two convolution layers. The numbers of filters are $64, 128, 256, 256, 256$, and 2 for the convolution layers, respectively. The kernel sizes are $(7, 7), (4, 4), (4, 4), (4, 4), (4, 4), (1, 1)$. And the strides are $(1, 1), (2, 2), (2, 2), (2, 2), (2, 2), (1, 1)$. The activation function of the first five layers is ReLU. The last convolution layer does not have an activation function. $G^o$ and $G^i$'s structures are almost the same. The only difference between them is the output size, 3 for $G^o$ and 128 for $G^i$. Each of them has two parts. The first part is made of 4 convolution layers, and there is an upsampling layer with a scale factor 2.0 between the second convolution layer and the third convolution layer. The numbers of filters are $128, 128, 64$, and 3 for the convolution layers, respectively. The kernel sizes are $(3, 3), (3, 3), (5, 5), (7, 7)$. The strides are $(1, 1)$ for all the convolution layers. The first and the third convolution layers' activation functions are ReLU. The fourth convolution layer's activation function is Tanh. The second convolution layer does not have an activation function. The second part is made of three fully connected layers. The number of neurons is 256 and 256, respectively, and the output size is 512. The activation function of the first two layers is ReLU, and there is no activation function on the output. $D^o$ comprises 4 convolution layers followed by an average pooling layer whose kernel size is 3, stride is 2, and padding is $[1, 1]$. The numbers of filters of the convolution layers are $64, 128, 256, 1$, respectively. The kernel sizes are $(4, 4)$ for the first three convolution layers and $(1, 1)$ for the fourth convolution layer. The strides are $(2, 2)$ for the first three convolution layers and $(1, 1)$ for the fourth convolution layer. The first three convolution layers' activation functions are LeakyReLU. The other layers do not have activation functions. $D^i$ is made of one fully connected layer whose input size is 112, and the output size is 64 with activation function ReLU. $h$ comprises one fully connected layer with input size 2, output size 1, and activation function Sigmoid. $f$ has two parts. The first part is Resnet-50 (He et al., 2016), and the second is one fully connected layer with input size 2048 and output size 2.

For the NYSF dataset: $E^m$ is made of two fully connected layers. The number of neurons is 32, and the output size is 16. The activation function of the first layer is ReLU, and there is no activation function on the output. $E^s$ is made of two fully connected layers. The number of neurons is 32, and the output size is 2. The activation function of the first layer is ReLU, and there is no activation function on the output. $G^o$ is made of two fully connected layers. The number of neurons is 32, and the output size is 51. The activation function of the first layer is ReLU, and there is no activation function on the output. $D^o$ is made of two fully connected layers. The number of neurons is 32, and the output size is 16. The activation function of the first layer is ReLU, and there is no activation function on the output. $E^c$ is made of two fully connected layers. The number of neurons is 16, and the output size is 8. The activation function of the first layer is ReLU, and there is no activation function on the output. $E^a$ is made of two fully connected layers. The number of neurons is 8, and the output size is 2. The activation function of the first layer is ReLU, and there is no activation function on the output. $G^i$ is made of two fully connected layers. The number of neurons is 16, and the output size is 16. The activation function of the first layer is ReLU, and there is no activation function on the output. $D^i$ is made of two fully connected layers. The number of neurons is 8, and the output size is 8. The activation function of the first layer is ReLU, and there is no activation function on the output. $h$ comprises one fully connected layer with input size 2 and output size 1. The activation function is Sigmoid. $f$ has two parts. The first part is made of 3 fully connected layers. The number of neurons is 32, and the output size is 32. The activation function of the first two layers is ReLU, and there is no activation function on the output. The second part is made of one fully connected layer whose input size is 32, the output size is 32, and it does not have an activation function.

## C.6  Hyperparameter Search

We follow the same set of the MUNIT (Huang et al., 2018) for the hyperparameters. More specifically, the learning rate is 0.0001, the number of iterations is 600000, and the batch size is 1. The loss weights in learning $T$ are chosen from $\{1, 5, 10\}$. The selected best ones are $\beta_d = 10, \beta_f = 1, \beta_z = 1, \beta_g = 1$. We monitor the loss of the validation set and choose the $\beta$ with the lowest validation loss.

For the hyperparameters in learning the classifier $f$, the learning rate is chosen from $\{0.000005, 0.00001, 0.00005, 0.0001, 0.0005\}$. $\eta$ is chosen from $\{0.01, 0.05, 0.1\}$. $\gamma$ is chosen from $\{0.01, 0.025, 0.05\}$. $\lambda$ is chosen from $\{0.1, 1, 10, 20\}$. The batch size is chosen from $\{22, 64, 80, 128, 512, 1024, 2048\}$. The numbers of iterations are chosen from $\{500, 1000, ..., 8000\}$ on the ccMNIST and NYSF datasets. The number of iterations are chosen from $\{300, 600, ..., 7800, 8000\}$ on the FairFace and YFCC100M-FDG datasets. The selected best ones are: the learning rate is 0.00005, $\eta_1 = \eta_2 = 0.05, \gamma_1 = \gamma_2 = 0.025, \lambda_1 = \lambda_2 = 1$. The batch size on the ccMNIST and YFCC100M-FDG datasets is 64, and it is 22 on the FairFace dataset and 1024 on the NYSF dataset. The number of iterations on the ccMNIST dataset is 3000, 500, 7000 for domains R, G, B, respectively. The number of iterations on the FairFace dataset is 7200, 7200, 7800, 8000, 6600, 7200, 6900 for domains B, E, I, L, M, S, W, respectively. The number of iterations on the YFCC100M-FDG dataset is 7200, 6000, 6900 for $d_0, d_1, d_2$, respectively. The number of iterations on the NYSF dataset is 500, 3500, 4000, 1500, 8000 for domains R, B, M, Q, S, respectively. We monitor the accuracy and the value of fairness metrics from the validation set and select the best ones. The grid space of the grid search on all the baselines is the same as for our method.

## C.7  Model Selection.

The model selection in domain generalization is intrinsically a learning problem, followed by (Robey et al., 2021), we use leave-one-domain-out validation criteria, which is one of the three selection methods stated in (Gulrajani & Lopez-Paz, 2020). Specifically, we evaluate FDDG on the held-out training domain and average the performance of $|\mathcal{E}_s| - 1$ domains over the held-out one.

# D  ABLATION STUDIES

We conduct three ablation studies, and detailed algorithms of designed ablation studies are given in Algorithms 3 to 5. For additional ablation study results on ccMNIST, YFCC100M-FDG, and NYSF, refer to Appendix F.

1. The difference between the full FDDG and the first ablation study (FDDG w/o sf) is that the latter does not have the inner level when learning $T$. Since the inner level is used to extract the content and sensitive factors from the semantic one, the same sensitive label of the generated images will remain due to the absence of $h(\cdot)$. Therefore, FDDG w/o sf is expected to have a lower level of fairness in the experiments. Results shown in the tables indicate that FDDG w/o sf has a significantly lower performance on fairness metrics.

2. The second study (FDDG w/o $T$) does not train the auto-encoders to generate images. All losses are computed only based on the sampled images. Similar to FDDG w/o sf, it is much harder to train a good classifier without the generated images in synthetic domains. Our results demonstrate that FDDG w/o $T$ performs worse on all the datasets.

3. The difference between FDDG and the third study (FDDG w/o fc) is that FDDG w/o fc does not have the fairness loss $\mathcal{L}_{fair}$ in line 9 of Algorithm 1. Therefore, this algorithm only focuses on accuracy without considering fairness. Results based on FDDG w/o fc show that it has a good level of accuracy but a poor level of fairness.

---

**Algorithm 3** FDDG w/o sf (Ablation Study 1)

---

1: **repeat**
2:     **for** minibatch $\mathcal{B} = \{(\mathbf{x}_i, z_i, y_i)\}_{i=1}^m \in \mathcal{D}_s$ **do**
3:         $\mathcal{L}_{cls}(\boldsymbol{\theta}) = (1/m)\sum_{i=1}^m \ell(y_i, \hat{f}(\mathbf{x}_i, \boldsymbol{\theta}))$
4:         $\mathcal{L}_{fair}(\boldsymbol{\theta}) = (1/m)\sum_{i=1}^m (\frac{1}{p_1(1-p_1)}(\frac{z_i+1}{2} - p_1)\hat{f}(\mathbf{x}_i, \boldsymbol{\theta})$
5:         **for** each $(\mathbf{x}_i, z_i, y_i)$ in the minibatch **do**
6:             $(\mathbf{x}'_i, z_i, y_i) = \text{DATAAUG}(\mathbf{x}_i, x_i, y_i)$
7:             $\mathcal{L}'_{cls}(\boldsymbol{\theta}) = (1/m)\sum_{i=1}^m \ell(y_i, \hat{f}(\mathbf{x}'_i, \boldsymbol{\theta}))$
8:         **end for**
9:         $\mathcal{L}_{cls}(\boldsymbol{\theta}) = \mathcal{L}_{cls}(\boldsymbol{\theta}) + \mathcal{L}'_{cls}(\boldsymbol{\theta})$
10:        $\mathcal{L}(\boldsymbol{\theta}) = \mathcal{L}_{cls}(\boldsymbol{\theta}) + \lambda_2 \cdot \mathcal{L}_{fair}(\boldsymbol{\theta})$
11:        $\boldsymbol{\theta} \leftarrow \boldsymbol{\theta} - \eta_p \cdot \nabla_{\boldsymbol{\theta}}\mathcal{L}(\boldsymbol{\theta})$
12:        $\lambda_2 \leftarrow \max\{[\lambda_2 + \eta_d \cdot (\mathcal{L}_{fair}(\boldsymbol{\theta}) - \gamma_2)], 0\}$
13:     **end for**
14: **until** convergence
15: **procedure** DATAAUG($\mathbf{x}, z, y$)
16:     $\mathbf{c} = E^m(\mathbf{x}, \boldsymbol{\theta}^m)$
17:     Sample $\mathbf{s}' \sim \mathcal{N}(0, I_s)$
18:     $\mathbf{x}' = G^o(\mathbf{c}, \mathbf{s}', \boldsymbol{\phi}_o)$
19:     **return** $(\mathbf{x}', z, y)$
20: **end procedure**

---

**Algorithm 4** FDDG w/o $T$ (Ablation Study 2)

---

1: **repeat**
2:     **for** minibatch $\mathcal{B} = \{(\mathbf{x}_i, z_i, y_i)\}_{i=1}^m \in \mathcal{D}_s$ **do**
3:         $\mathcal{L}_{cls}(\boldsymbol{\theta}) = (1/m)\sum_{i=1}^m \ell(y_i, \hat{f}(\mathbf{x}_i, \boldsymbol{\theta}))$
4:         $\mathcal{L}_{fair}(\boldsymbol{\theta}) = (1/m)\sum_{i=1}^m (\frac{1}{p_1(1-p_1)}(\frac{z_i+1}{2} - p_1)\hat{f}(\mathbf{x}_i, \boldsymbol{\theta})$
5:         $\mathcal{L}(\boldsymbol{\theta}) = \mathcal{L}_{cls}(\boldsymbol{\theta}) + \lambda_2 \cdot \mathcal{L}_{fair}(\boldsymbol{\theta})$
6:         $\boldsymbol{\theta} \leftarrow \boldsymbol{\theta} - \eta_p \cdot \nabla_{\boldsymbol{\theta}}\mathcal{L}(\boldsymbol{\theta})$
7:         $\lambda_2 \leftarrow \max\{[\lambda_2 + \eta_d \cdot (\mathcal{L}_{fair}(\boldsymbol{\theta}) - \gamma_2)], 0\}$
8:     **end for**
9: **until** convergence

---

---

**Algorithm 5** FDDG w/o fc (Ablation Study 3)

---

1: **repeat**
2:     **for** minibatch $\mathcal{B} = \{(\mathbf{x}_i, z_i, y_i)\}_{i=1}^m \in \mathcal{D}_s$ **do**
3:         $\mathcal{L}_{cls}(\boldsymbol{\theta}) = (1/m) \sum_{i=1}^m \ell(y_i, \hat{f}(\mathbf{x}_i, \boldsymbol{\theta}))$
4:         Initialize $\mathcal{L}'_{inv}(\boldsymbol{\theta}) = 0$
5:         **for** each $(\mathbf{x}_i, z_i, y_i)$ in the minibatch **do**
6:             $(\mathbf{x}'_i, y_i) = \text{DATAAUG}(\mathbf{x}_i, z_i, y_i)$
7:             $\mathcal{L}'_{inv}(\boldsymbol{\theta}) += d[\hat{f}(\mathbf{x}_i, \boldsymbol{\theta}), \hat{f}(\mathbf{x}'_i, \boldsymbol{\theta})]$
8:         **end for**
9:         $\mathcal{L}_{inv}(\boldsymbol{\theta}) = \mathcal{L}'_{inv}(\boldsymbol{\theta})/m$
10:         $\mathcal{L}(\boldsymbol{\theta}) = \mathcal{L}_{cls}(\boldsymbol{\theta}) + \lambda_1 \cdot \mathcal{L}_{inv}(\boldsymbol{\theta})$
11:         $\boldsymbol{\theta} \leftarrow \boldsymbol{\theta} - \eta_p \cdot \nabla_{\boldsymbol{\theta}} \mathcal{L}(\boldsymbol{\theta})$
12:         $\lambda_1 \leftarrow \max\{[\lambda_1 + \eta_d \cdot (\mathcal{L}_{inv}(\boldsymbol{\theta}) - \gamma_1)], 0\}$
13:     **end for**
14: **until** convergence
15: **procedure** DATAAUG$(\mathbf{x}, z, y)$
16:     $\mathbf{c} = E^c(E^m(\mathbf{x}, \boldsymbol{\theta}^m), \boldsymbol{\theta}^c)$
17:     Sample $\mathbf{a}' \sim \mathcal{N}(0, I_a)$
18:     Sample $\mathbf{s}' \sim \mathcal{N}(0, I_s)$
19:     $\mathbf{x}' = G^o(G^i(\mathbf{c}, \mathbf{a}', \boldsymbol{\phi}_i), \mathbf{s}', \boldsymbol{\phi}_o)$
20:     **return** $(\mathbf{x}', z, y)$
21: **end procedure**

---

# E   PROOFS

## E.1   SKETCH PROOF OF THEOREM 1

**Lemma 1.** *Given two domains $e_i, e_j \in \mathcal{E}$, $\mathbb{E}_{\mathbb{P}(X^{e_j}, Z^{e_j})} g(f(X^{e_j}), Z^{e_j})$ can be bounded by $\mathbb{E}_{\mathbb{P}(X^{e_i}, Z^{e_i})} g(f(X^{e_i}), Z^{e_i})$ as follows:*

$$\mathbb{E}_{\mathbb{P}(X^{e_j}, Z^{e_j})} g(f(X^{e_j}), Z^{e_j}) \leq \mathbb{E}_{\mathbb{P}(X^{e_i}, Z^{e_i})} g(f(X^{e_i}), Z^{e_i}) + \sqrt{2} D[\mathbb{P}(X^{e_j}, Z^{e_j}, Y^{e_j}), \mathbb{P}(X^{e_i}, Z^{e_i}, Y^{e_i})]$$

**Lemma 2.** *Given two domains $e_i, e_j \in \mathcal{E}$, under Lemma 1, $\epsilon^{e_j}(f)$ can be bounded by $\epsilon^{e_i}(f)$ as follows:*

$$\epsilon^{e_j}(f) \leq \epsilon^{e_i}(f) + \sqrt{2} D[\mathbb{P}(X^{e_j}, Z^{e_j}, Y^{e_j}), \mathbb{P}(X^{e_i}, Z^{e_i}, Y^{e_i})]$$

Under Lemmas 1 and 2, we now prove Theorem 1

*Proof.* Let $e_\star \in \mathcal{E}_s$ be the source domain nearest to the target domain $e_t \in \mathcal{E} \backslash \mathcal{E}_s$. Under Lemma 2, we have

$$\epsilon^{e_t}(f) \leq \epsilon^{e_i}(f) + \sqrt{2} D[\mathbb{P}(X^{e_t}, Z^{e_t}, Y^{e_t}), \mathbb{P}(X^{e_i}, Z^{e_i}, Y^{e_i})]$$

where $e_i \in \mathcal{E}_s$. Taking average of upper bounds based on all source domains, we have:

$$\epsilon^{e_t}(f) \leq \frac{1}{|\mathcal{E}_s|} \sum_{e_i \in \mathcal{E}_s} \epsilon^{e_i}(f) + \frac{\sqrt{2}}{|\mathcal{E}_s|} \sum_{e_i \in \mathcal{E}_s} D[\mathbb{P}(X^{e_t}, Z^{e_t}, Y^{e_t}), \mathbb{P}(X^{e_i}, Z^{e_i}, Y^{e_i})]$$

$$\leq \frac{1}{|\mathcal{E}_s|} \sum_{e_i \in \mathcal{E}_s} \epsilon^{e_i}(f) + \frac{\sqrt{2}}{|\mathcal{E}_s|} |\mathcal{E}_s| D[\mathbb{P}(X^{e_t}, Z^{e_t}, Y^{e_t}), \mathbb{P}(X^{e_\star}, Z^{e_\star}, Y^{e_\star})]$$

$$+ \frac{\sqrt{2}}{|\mathcal{E}_s|} \sum_{e_i \in \mathcal{E}_s} D[\mathbb{P}(X^{e_\star}, Z^{e_\star}, Y^{e_\star}), \mathbb{P}(X^{e_i}, Z^{e_i}, Y^{e_i})]$$

$$\leq \frac{1}{|\mathcal{E}_s|} \sum_{e_i \in \mathcal{E}_s} \epsilon^{e_i}(f) + \sqrt{2} \min_{e_i \in \mathcal{E}_s} D[\mathbb{P}(X^{e_t}, Z^{e_t}, Y^{e_t}), \mathbb{P}(X^{e_i}, Z^{e_i}, Y^{e_i})]$$

$$+ \sqrt{2} \max_{e_i, e_j \in \mathcal{E}_s} D[\mathbb{P}(X^{e_i}, Z^{e_i}, Y^{e_i}), \mathbb{P}(X^{e_j}, Z^{e_j}, Y^{e_j})]$$

$\square$

### E.2 Sketch Proof of Theorem 2

Before we prove Theorem 2, we first make the following propositions and assumptions.

**Proposition 1.** *Let $d$ be a distance metric between probability measures for which it holds that $d[\mathbb{P},\mathbb{T}] = 0$ for two distributions $\mathbb{P}$ and $\mathbb{T}$ if and only if $\mathbb{P} = \mathbb{T}$ almost surely. Then $P^\star(0,0) = P^\star$*

**Proposition 2.** *Assuming the perturbation function $P^\star(\gamma_1,\gamma_2)$ is $L$-lipschitz continuous in $\gamma_1,\gamma_2$. Then given Proposition 1, it follows that $|P^\star - P^\star(\gamma_1,\gamma_2)| \leq L\|\boldsymbol{\gamma}\|_1$, where $\boldsymbol{\gamma} = [\gamma_1,\gamma_2]^T$.*

**Definition 4.** *Let $\Theta \subseteq \mathbb{R}^p$ be a finite-dimensional parameter space. For $\xi > 0$, a function $\hat{f} : \mathcal{X}\times\Theta\to \mathcal{Y}$ is said to be an $\xi$-parameterization of $\mathcal{F}$ if it holds that for each $f \in \mathcal{F}$, there exists a parameter $\boldsymbol{\theta} \in \Theta$ such that $\mathbb{E}_{\mathbb{P}(X)}\|\hat{f}(\mathbf{x},\boldsymbol{\theta}) - f(\mathbf{x})\|_\infty \leq \xi$. Given an $\xi$-parameterization $\hat{f}$ of $\mathcal{F}$, consider the following saddle-point problem:*

$$D_\xi^\star(\gamma_1,\gamma_2) \triangleq \max_{\lambda_1(e_i,e_j),\lambda_2(e_i,e_j)} \min_{\boldsymbol{\theta}\in\Theta} R(\boldsymbol{\theta}) + \int_{e_i,e_j\in\mathcal{E}} [\delta^{e_i,e_j}(\boldsymbol{\theta}) - \gamma_1]\mathrm{d}\lambda_1(e_i,e_j)$$
$$+ \int_{e_i,e_j\in\mathcal{E}} [\epsilon^{e_i}(\boldsymbol{\theta}) + \epsilon^{e_j}(\boldsymbol{\theta}) - \gamma_2]\mathrm{d}\lambda_2(e_i,e_j)$$

*where $R(\boldsymbol{\theta}) = R(\hat{f}(\cdot,\boldsymbol{\theta}))$ and $\mathcal{L}^{e_i,e_j}(\boldsymbol{\theta}) = \mathcal{L}^{e_i,e_j}(\hat{f}(\cdot,\boldsymbol{\theta}))$.*

**Assumption 4.** *The loss function $\ell$ is non-negative, convex, and $L_\ell$-Lipschitz continuous in its first argument,*

$$|\ell(f_1(\mathbf{x}),y) - \ell(f_2(\mathbf{x}),y)| \leq \|f_1(\mathbf{x}) - f_2(\mathbf{x})\|_\infty$$

**Assumption 5.** *The distance metric $d$ is non-negative, convex, and satisfies the following uniform Lipschitz-like inequality for some constant $L_d > 0$:*

$$|d[f_1(\mathbf{x}),f_1(\mathbf{x}' = T(\mathbf{x},z,e))] - d[f_2(\mathbf{x}),f_2(\mathbf{x}' = T(\mathbf{x},z,e))]| \leq L_d\|f_1(\mathbf{x}) - f_2(\mathbf{x})\|_\infty, \quad \forall e \in \mathcal{E}$$

**Assumption 6.** *The fairness metric $g$ is non-negative, convex, and satisfies the following uniform Lipschitz-like inequality for some constant $L_g > 0$:*

$$|(g\circ f_1)(\mathbf{x},z) - (g\circ f_2)(\mathbf{x},z)| \leq L_g\|f_1(\mathbf{x}) - f_2(\mathbf{x})\|_\infty, \quad \forall e \in \mathcal{E}$$

**Assumption 7.** *There exists a parameter $\boldsymbol{\theta} \in \Theta$ such that $\delta^{e_i,e_j}(\boldsymbol{\theta}) < \gamma_1 - \xi \cdot \max\{L_\ell, L_d\}$ and $\epsilon^{e_i}(\boldsymbol{\theta}) + \epsilon^{e_j}(\boldsymbol{\theta}) < \gamma_2 - \xi \cdot \max\{L_\ell, L_g\}, \forall e_i,e_j \in \mathcal{E}$*

**Proposition 3.** *Let $\gamma_1,\gamma_2 > 0$ be given. With the assumptions above, it holds that*

$$P^\star(\gamma_1,\gamma_2) \leq D_\xi^\star(\gamma_1,\gamma_2) \leq P^\star(\gamma_1,\gamma_2) + \xi(1 + \|\lambda_p^\star\|_1)\cdot k$$

*where $\lambda_p^\star$ is the optimal dual variable for a perturbed version of Eq. (5) in which the constraints are tightened to hold with margin $\gamma - \xi \cdot k$, $k = \max\{L_\ell, L_d, L_g\}$. In particular, this result implies that*

$$|P^\star(\gamma_1,\gamma_2) - D_\xi^\star(\gamma_1,\gamma_2)| \leq \xi k(1 + \|\lambda_p^\star\|_{L_1})$$

**Proposition 4** (Empirical gap)**.** *Assume $\ell$ and $d$ are non-negative and bounded in $[-B, B]$ and let $d_{\mathrm{VC}}$ denote the VC-dimension of the hypothesis class $\mathcal{A}_\xi = \{\hat{f}(\cdot,\boldsymbol{\theta}) : \boldsymbol{\theta} \in \Theta\} \subseteq \mathcal{F}$. Then it holds with probability $1 - \omega$ over the $N$ samples from each domain that*

$$|D_\xi^\star(\gamma_1,\gamma_2) - D_{\xi,N,\mathcal{E}_s}^\star(\gamma_1,\gamma_2)| \leq 2B\sqrt{\frac{1}{N}[1 + \log(\frac{4(2N)^{d_{\mathrm{VC}}}}{\omega})]}$$

The Theorem 2. Let $\xi > 0$ be given, and let $\hat{f}$ be an $\xi$-parameterization of $\mathcal{F}$. Let the assumptions holds, and further assume that $\ell$, $d$, and $g$ are $[0, B]$-bounded and that $d[\mathbb{P},\mathbb{T}] = 0$ if and only if $\mathbb{P} = \mathbb{T}$ almost surely, and that $P^\star(\gamma_1,\gamma_2)$ is $L$-Lipschitz. Then assuming that $\mathcal{A}_\xi = \{\hat{f}(\cdot,\theta) : \theta \in \Theta\} \subseteq \mathcal{F}$ has finite VC-dimension, it holds with probability $1 - \omega$ over the $N$ samples that

$$|P^\star - D_{\xi,N,\mathcal{E}_s}^\star(\boldsymbol{\gamma})| \leq L\|\boldsymbol{\gamma}\|_1 + \xi k(1 + \|\boldsymbol{\lambda}_p^\star\|_1) + O(\sqrt{\log(M)/M})$$

Now we prove Theorem 2.

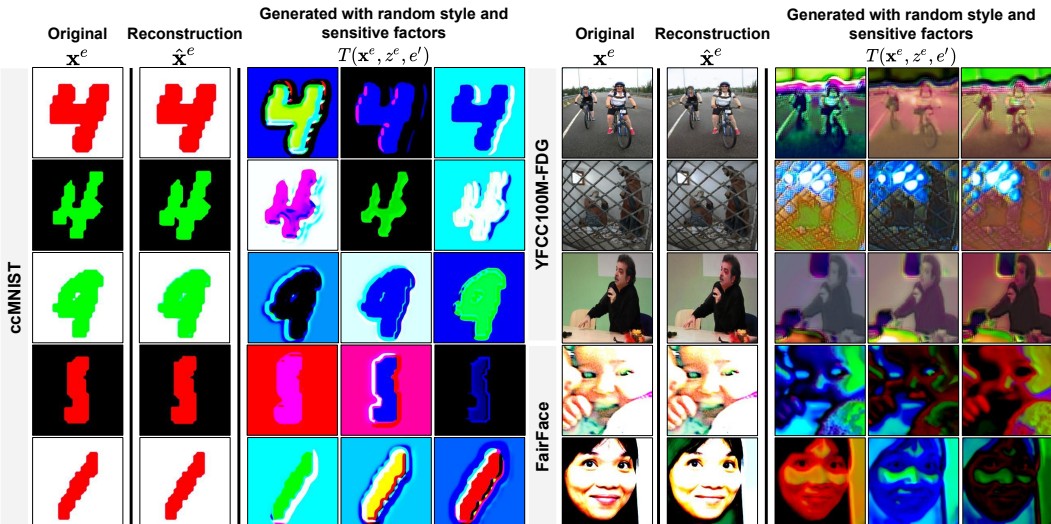

Figure 6: Additional visualizations for data reconstruction and under the transformation model $T$.

Table 6: Full performance on ccMNIST. (bold is the best; underline is the second best).

| Methods | DP ↑ / AUC ↓ / Accuracy ↑ | | | |
| | (R, 0.11) | (G, 0.43) | (B, 0.87) | Avg |
| --- | --- | --- | --- | --- |
| RandAug | 0.11±0.05 / 0.95±0.02 / 90.59±0.23 | 0.44±0.01 / 0.71±0.03 / 87.62±0.22 | 0.87±0.03 / 0.66±0.01 / 86.33±1.50 | 0.47 / 0.77 / 88.18 |
| ERM | 0.12±0.25 / 0.91±0.03 / 98.00±1.14 | 0.43±0.23 / 0.78±0.01 / 98.07±0.35 | 0.89±0.06 / 0.64±0.01 / 95.64±1.75 | 0.48 / 0.78 / 97.24 |
| IRM | 0.21±0.15 / 0.97±0.02 / 75.50±2.11 | 0.28±0.10 / 0.64±0.01 / 92.74±0.27 | 0.76±0.12 / 0.63±0.03 / 80.05±2.34 | 0.42 / 0.75 / 82.76 |
| GDRO | 0.12±0.09 / 0.92±0.03 / 98.19±0.93 | 0.43±0.06 / 0.75±0.03 / 98.17±0.87 | 0.90±0.07 / 0.65±0.01 / 95.03±0.12 | 0.48 / 0.77 / 97.13 |
| Mixup | 0.12±0.21 / 0.92±0.02 / 97.89±1.97 | 0.41±0.06 / 0.79±0.02 / 98.00±1.36 | 0.93±0.04 / 0.65±0.01 / 96.09±1.07 | 0.49 / 0.79 / 97.32 |
| MLDG | 0.11±0.12 / 0.91±0.03 / 98.52±0.94 | 0.43±0.22 / 0.77±0.02 / **98.67**±0.61 | 0.87±0.09 / 0.62±0.03 / 93.76±1.50 | 0.46 / 0.77 / 96.98 |
| CORAL | 0.11±0.08 / 0.91±0.03 / **98.69**±0.76 | 0.42±0.20 / 0.79±0.02 / 98.30±0.98 | 0.87±0.07 / 0.64±0.01 / 93.74±1.54 | 0.47 / 0.78 / 96.91 |
| MMD | 0.11±0.08 / 0.92±0.01 / **98.69**±1.07 | 0.41±0.21 / 0.73±0.03 / 97.72±1.31 | 0.93±0.04 / 0.59±0.01 / 95.37±1.56 | 0.48 / 0.75 / 97.26 |
| DANN | 0.14±0.08 / 0.87±0.03 / 85.94±1.76 | 0.17±0.13 / 0.90±0.03 / 84.93±0.67 | 0.76±0.17 / 0.63±0.03 / 84.04±1.75 | 0.36 / 0.80 / 84.97 |
| CDANN | 0.19±0.13 / 0.90±0.01 / 93.03±2.18 | 0.60±0.17 / 0.89±0.03 / 71.92±1.03 | 0.77±0.14 / 0.63±0.02 / 84.03±1.96 | 0.52 / 0.81 / 82.99 |
| DDG | 0.11±0.07 / 0.91±0.01 / 98.26±2.38 | 0.42±0.14 / 0.77±0.02 / 98.14±0.11 | **0.96**±0.03 / 0.60±0.01 / **97.02**±1.70 | 0.50 / 0.76 / **97.81** |
| MBDG | 0.12±0.04 / 0.93±0.01 / 98.47±0.94 | 0.42±0.08 / 0.81±0.03 / 97.62±1.87 | 0.90±0.08 / 0.64±0.03 / 96.01±2.26 | 0.48 / 0.79 / 97.37 |
| DDG-FC | 0.11±0.04 / 0.91±0.03 / 96.69±1.12 | 0.42±0.05 / 0.75±0.01 / 96.09±1.86 | **0.97**±0.02 / 0.58±0.01 / 95.66±2.17 | 0.50 / 0.75 / 96.14 |
| MBDG-FC | 0.13±0.08 / 0.91±0.02 / 98.07±1.06 | 0.45±0.20 / 0.76±0.03 / 96.09±0.61 | 0.94±0.04 / 0.64±0.01 / 95.42±1.13 | 0.50 / 0.77 / 96.52 |
| EIIL | 0.15±0.08 / 0.94±0.03 / 81.00±0.31 | 0.26±0.06 / 0.98±0.01 / 82.67±2.44 | 0.62±0.16 / 0.98±0.01 / 71.68±0.51 | 0.34 / 0.97 / 78.45 |
| FarconVAE | 0.11±0.08 / 0.94±0.01 / 94.40±2.35 | 0.43±0.21 / 0.77±0.03 / 82.61±1.90 | **0.97**±0.02 / 0.59±0.01 / 76.22±0.45 | 0.50 / 0.77 / 84.41 |
| FATDM | 0.17±0.03 / 0.86±0.02 / 96.00±0.23 | 0.92±0.02 / 0.64±0.01 / 95.55±1.10 | 0.90±0.06 / **0.57**±0.03 / 95.23±0.55 | 0.66 / 0.67 / 95.59 |
| FDDG | **0.23**±0.09 / **0.84**±0.01 / 96.15±0.50 | **0.98**±0.01 / **0.58**±0.01 / 97.94±0.30 | 0.92±0.05 / **0.57**±0.03 / 96.19±1.33 | **0.71** / **0.66** / 96.76 |

*Proof.* The proof of this theorem is a simple consequence of the triangle inequality. Indeed, by combining Proposition 2, Proposition 3, and Proposition 4, we find that

$$
\begin{aligned}
&|P^\star - D^\star_{\xi,N,\mathcal{E}_s}(\gamma_1,\gamma_2)| \\
=&|P^\star + P^\star(\gamma_1,\gamma_2) - P^\star(\gamma_1,\gamma_2) + D^\star_\xi(\gamma_1,\gamma_2) - D^\star_\xi(\gamma_1,\gamma_2) - D^\star_{\xi,N,\mathcal{E}_s}(\gamma_1,\gamma_2)| \\
\leq&|P^\star - P^\star(\gamma_1,\gamma_2)| + |P^\star(\gamma_1,\gamma_2) - D^\star_\xi(\gamma_1,\gamma_2)| + |D^\star_\xi(\gamma_1,\gamma_2) - D^\star_{\xi,N,\mathcal{E}_s}(\gamma_1,\gamma_2)| \\
\leq& L\|\gamma\|_1 + \xi k(1 + \|\lambda^\star_p\|_1) + 2B\sqrt{\frac{1}{N}[1 + \log(\frac{4(2N)^{d_{\mathrm{VC}}}}{\omega})]}
\end{aligned}
$$

□

# F ADDITIONAL RESULTS

Additional results including more visualization (Fig. 6), as well as complete results with all domains and baselines on ccMNIST (Tab. 6), FairFace (Tab. 7), FairFace (Tab. 8), and NYSF (Tab. 9) are provided. Additional ablation study results are in Tabs. 10 to 13.

**Trade-off between fairness-accuracy.** In our algorithm, because $\lambda_2$ is the parameter that regularizes the fair loss, we conduct additional experiments to show the change between accuracy and

Table 7: Full performance on `FairFace`. (bold is the best; underline is the second best).

| Methods | DP ↑ / AUC ↓ / Accuracy ↑ | | |
|---|---|---|---|
| | (B, 0.91) | (E, 0.87) | (I, 0.58) |
| RandAug | 0.64±0.26 / 0.64±0.15 / 93.47±1.56 | 0.41±0.34 / 0.68±0.09 / 95.62±1.96 | 0.44±0.21 / 0.63±0.05 / 92.99±1.00 |
| ERM | 0.67±0.17 / 0.58±0.02 / 91.89±1.10 | 0.43±0.21 / 0.64±0.02 / 95.69±2.19 | 0.50±0.19 / 0.59±0.03 / 93.28±1.61 |
| IRM | 0.63±0.12 / 0.58±0.01 / 93.39±1.03 | 0.32±0.23 / 0.63±0.03 / 95.12±0.49 | 0.45±0.06 / 0.59±0.02 / 92.01±1.13 |
| GDRO | 0.71±0.16 / 0.57±0.02 / 89.81±1.10 | 0.46±0.16 / 0.61±0.02 / 95.26±1.53 | 0.50±0.14 / 0.59±0.01 / 93.27±1.27 |
| Mixup | 0.58±0.19 / 0.59±0.02 / 92.46±0.69 | 0.40±0.04 / 0.61±0.02 / 93.31±1.42 | 0.42±0.09 / 0.59±0.02 / **93.42**±2.43 |
| MLDG | 0.63±0.25 / 0.58±0.02 / 92.71±2.36 | 0.41±0.15 / 0.62±0.03 / 95.59±0.87 | 0.51±0.15 / 0.60±0.02 / 93.35±1.87 |
| CORAL | 0.69±0.19 / 0.58±0.01 / 92.09±2.03 | 0.34±0.24 / 0.64±0.01 / **95.91**±1.44 | 0.53±0.05 / 0.59±0.02 / 93.35±0.26 |
| MMD | 0.69±0.25 / 0.56±0.01 / 93.87±0.14 | 0.45±0.22 / **0.57**±0.02 / 94.68±0.20 | 0.27±0.18 / **0.57**±0.03 / 89.88±0.22 |
| DANN | 0.46±0.07 / 0.61±0.02 / 91.80±0.64 | 0.53±0.18 / 0.85±0.03 / 91.54±2.24 | 0.38±0.18 / 0.63±0.01 / 90.09±0.60 |
| CDANN | 0.62±0.24 / 0.59±0.03 / 91.22±0.33 | 0.43±0.10 / 0.66±0.02 / 94.75±2.23 | 0.43±0.18 / 0.61±0.01 / 92.41±1.68 |
| DDG | 0.60±0.20 / 0.59±0.02 / 91.76±1.03 | 0.36±0.15 / 0.63±0.02 / 95.52±2.35 | 0.49±0.17 / 0.59±0.01 / 92.35±2.04 |
| MBDG | 0.60±0.15 / 0.58±0.01 / 91.29±1.41 | 0.46±0.10 / 0.63±0.01 / 95.01±1.39 | 0.52±0.14 / 0.58±0.02 / 92.77±2.07 |
| DDG-FC | 0.61±0.06 / 0.58±0.03 / 92.27±1.65 | 0.39±0.18 / 0.64±0.01 / 95.51±2.36 | 0.45±0.17 / 0.58±0.03 / 93.38±0.52 |
| MBDG-FC | 0.70±0.15 / 0.56±0.03 / 92.12±0.43 | 0.35±0.07 / 0.60±0.01 / 95.54±1.80 | **0.56**±0.07 / **0.57**±0.01 / 92.41±1.61 |
| EIIL | 0.88±0.07 / 0.59±0.05 / 84.75±2.16 | 0.69±0.12 / 0.71±0.01 / 92.86±1.70 | 0.47±0.08 / **0.57**±0.01 / 86.93±0.89 |
| FarconVAE | 0.93±0.03 / **0.54**±0.01 / 89.61±0.64 | 0.72±0.17 / 0.63±0.01 / 91.50±1.89 | 0.42±0.24 / 0.58±0.03 / 87.42±2.14 |
| FATDM | 0.93±0.03 / 0.57±0.02 / 92.20±0.36 | 0.80±0.02 / 0.65±0.02 / 92.89±1.00 | 0.52±0.10 / 0.60±0.01 / 92.22±1.60 |
| FDDG | **0.94**±0.05 / 0.55±0.02 / **93.91**±0.33 | **0.87**±0.05 / 0.60±0.01 / **95.91**±1.06 | 0.48±0.06 / **0.57**±0.02 / 92.55±1.45 |

| Methods | DP ↑ / AUC ↓ / Accuracy ↑ | | |
|---|---|---|---|
| | (M, 0.87) | (S, 0.39) | (W, 0.49) |
| RandAug | 0.36±0.12 / 0.65±0.05 / 92.79±1.22 | 0.35±0.20 / 0.69±0.06 / 91.89±1.02 | 0.34±0.09 / 0.64±0.02 / 92.07±0.55 |
| ERM | 0.34±0.08 / 0.62±0.01 / 92.51±1.45 | 0.68±0.14 / 0.59±0.03 / 93.48±0.94 | 0.39±0.09 / 0.61±0.01 / 92.82±0.38 |
| IRM | 0.34±0.11 / 0.65±0.02 / 92.47±2.42 | 0.55±0.23 / 0.59±0.01 / 91.81±0.66 | 0.32±0.19 / 0.66±0.01 / 90.54±1.56 |
| GDRO | 0.45±0.14 / 0.63±0.02 / 91.75±1.11 | 0.72±0.14 / 0.59±0.01 / 93.65±0.67 | 0.48±0.09 / 0.60±0.01 / 92.50±0.38 |
| Mixup | 0.31±0.11 / 0.62±0.02 / **93.52**±0.79 | 0.91±0.04 / 0.58±0.02 / 93.20±0.33 | 0.43±0.19 / 0.61±0.01 / 92.98±0.03 |
| MLDG | 0.35±0.20 / 0.62±0.01 / 92.45±0.07 | 0.71±0.22 / 0.57±0.01 / **93.85**±0.40 | 0.47±0.20 / 0.59±0.01 / 92.82±1.65 |
| CORAL | 0.43±0.08 / 0.63±0.01 / 92.23±0.06 | 0.74±0.10 / 0.58±0.01 / 93.77±1.99 | 0.50±0.14 / 0.60±0.02 / 92.47±2.04 |
| MMD | 0.48±0.25 / 0.62±0.02 / 91.07±2.00 | 0.66±0.18 / 0.59±0.03 / 92.58±1.63 | 0.39±0.20 / 0.68±0.02 / 91.75±1.37 |
| DANN | **0.65**±0.14 / 0.88±0.01 / 91.46±0.50 | 0.80±0.14 / 0.57±0.02 / 88.20±1.65 | 0.11±0.09 / 0.66±0.01 / 86.80±1.18 |
| CDANN | 0.27±0.12 / 0.67±0.01 / 91.07±0.97 | 0.52±0.12 / 0.82±0.02 / 88.32±0.37 | 0.35±0.17 / 0.67±0.02 / 90.19±0.60 |
| DDG | 0.37±0.14 / 0.64±0.01 / 91.36±0.65 | 0.63±0.22 / 0.58±0.01 / 93.40±0.37 | 0.51±0.07 / 0.60±0.01 / 91.34±0.80 |
| MBDG | 0.38±0.14 / 0.64±0.02 / 92.23±1.15 | 0.67±0.06 / 0.56±0.03 / 93.12±0.70 | 0.30±0.04 / 0.62±0.01 / 91.05±0.53 |
| DDG-FC | 0.42±0.09 / 0.95±0.03 / 92.70±1.49 | 0.76±0.21 / 0.59±0.02 / **93.85**±1.79 | 0.48±0.15 / 0.62±0.02 / 92.45±1.55 |
| MBDG-FC | 0.49±0.19 / 0.63±0.03 / 90.67±0.42 | 0.74±0.23 / 0.57±0.01 / 93.24±0.32 | 0.32±0.07 / 0.60±0.03 / 91.50±0.57 |
| EIIL | 0.52±0.09 / 0.63±0.03 / 84.96±1.37 | **0.98**±0.01 / **0.55**±0.02 / 89.99±2.27 | 0.46±0.05 / 0.65±0.03 / 86.53±1.02 |
| FarconVAE | 0.54±0.22 / **0.58**±0.02 / 85.62±1.49 | 0.92±0.06 / 0.56±0.10 / 90.00±0.05 | 0.51±0.07 / 0.60±0.01 / 86.40±0.42 |
| FATDM | 0.55±0.12 / 0.65±0.01 / 92.23±1.56 | 0.92±0.10 / 0.57±0.02 / 92.36±0.99 | 0.46±0.05 / 0.63±0.01 / 92.56±0.31 |
| FDDG | 0.54±0.08 / 0.62±0.02 / 92.61±1.84 | **0.98**±0.01 / **0.55**±0.01 / 92.26±2.48 | **0.52**±0.17 / **0.58**±0.03 / **93.02**±0.50 |

| Methods | DP ↑ / AUC ↓ / Accuracy ↑ | |
|---|---|---|
| | (L, 0.48) | Avg |
| RandAug | 0.39±0.10 / 0.70±0.02 / 91.77±0.61 | 0.42 / 0.66 / 92.94 |
| ERM | 0.57±0.15 / 0.62±0.01 / 91.96±0.51 | 0.51 / 0.61 / 93.08 |
| IRM | 0.41±.021 / 0.63±0.05 / 92.06±1.89 | 0.43 / 0.62 / 92.48 |
| GDRO | 0.54±0.15 / 0.62±0.01 / 91.59±0.51 | 0.55 / 0.60 / 92.55 |
| Mixup | 0.55±0.22 / 0.61±0.02 / 93.43±2.02 | 0.51 / 0.60 / 93.19 |
| MLDG | 0.53±0.18 / 0.62±0.03 / 92.99±0.86 | 0.51 / 0.60 / 93.39 |
| CORAL | 0.56±0.23 / **0.59**±0.03 / 92.62±1.11 | 0.54 / 0.60 / 93.21 |
| MMD | 0.55±0.16 / 0.61±0.02 / 92.53±1.41 | 0.50 / 0.60 / 92.34 |
| DANN | 0.39±0.21 / 0.67±0.01 / 90.82±2.44 | 0.47 / 0.70 / 90.10 |
| CDANN | 0.42±0.23 / 0.61±0.03 / 92.42±2.19 | 0.43 / 0.66 / 91.48 |
| DDG | 0.44±0.17 / 0.62±0.02 / 93.46±0.32 | 0.49 / 0.61 / 92.74 |
| MBDG | 0.56±0.09 / 0.61±0.01 / 93.49±0.97 | 0.50 / 0.60 / 92.71 |
| DDG-FC | 0.50±0.25 / 0.62±0.03 / 92.42±0.30 | 0.52 / 0.61 / 93.23 |
| MBDG-FC | 0.57±0.23 / 0.62±0.02 / 91.89±0.81 | 0.53 / 0.60 / 92.48 |
| EIIL | 0.49±0.07 / **0.59**±0.01 / 88.39±1.25 | 0.64 / 0.61 / 87.78 |
| FarconVAE | **0.58**±0.05 / 0.60±0.05 / 88.70±0.71 | 0.66 / **0.58** / 88.46 |
| FATDM | 0.51±0.16 / 0.63±0.02 / 93.33±0.20 | 0.67 / 0.61 / 92.54 |
| FDDG | **0.58**±0.15 / **0.59**±0.01 / **93.73**±0.26 | **0.70** / **0.58** / **93.42** |

fairness. Our results show that the larger (smaller) $\lambda_2$, the better (worse) model fairness for each domain as well as in average, but it gives worse (better) model utility. Moreover, we show additional experiment results based on choosing different $\gamma_1$ and $\gamma_2$. We observe that (1) by only increasing $\gamma_2$, the model towards giving unfair outcomes but higher accuracy; (2) by only increasing $\gamma_1$, performance on both model fairness and accuracy decreases. This may be due to the failure of disentanglement of factors. Evaluation on all datasets of fairness-accuracy trade-offs is given in Table 14. Results in the table are average performance over target domains.

Table 8: Full performance on `YFCC100M-FDG`. (bold is the best; underline is the second best).

| Methods | DP ↑ / AUC ↓ / Accuracy ↑ | | | |
| --- | --- | --- | --- | --- |
| | $(d_0, 0.73)$ | $(d_1, 0.84)$ | $(d_2, 0.72)$ | **Avg** |
| RandAug | 0.67±0.06 / 0.57±0.02 / 57.47±1.20 | 0.67±0.34 / 0.61±0.01 / 82.43±1.25 | 0.65±0.21 / 0.64±0.02 / 87.88±0.35 | 0.66 / 0.61 / 75.93 |
| ERM | 0.81±0.09 / 0.58±0.01 / 40.51±0.23 | 0.71±0.18 / 0.66±0.03 / 83.91±0.33 | 0.89±0.08 / 0.59±0.01 / 82.06±0.33 | 0.80 / 0.61 / 68.83 |
| IRM | 0.76±0.10 / 0.58±0.02 / 50.51±2.44 | 0.87±0.19 / 0.60±0.02 / 73.26±0.03 | 0.70±0.24 / 0.57±0.02 / 82.78±2.19 | 0.78 / 0.58 / 68.85 |
| GDRO | 0.80±0.05 / 0.59±0.01 / 53.43±2.29 | 0.73±0.22 / 0.60±0.01 / 87.56±2.20 | 0.79±0.13 / 0.65±0.02 / 83.10±0.64 | 0.78 / 0.62 / 74.70 |
| Mixup | 0.82±0.07 / 0.57±0.03 / 61.15±0.28 | 0.79±0.14 / 0.63±0.03 / 78.63±0.97 | 0.89±0.05 / 0.60±0.01 / 85.18±0.80 | 0.84 / 0.60 / 74.99 |
| MLDG | 0.75±0.13 / 0.67±0.01 / 49.56±0.69 | 0.71±0.19 / 0.57±0.02 / 89.45±0.44 | 0.71±0.14 / 0.57±0.03 / 87.51±0.18 | 0.72 / 0.60 / 75.51 |
| CORAL | 0.80±0.11 / 0.58±0.02 / 58.96±2.34 | 0.72±0.11 / 0.64±0.03 / 91.66±0.85 | 0.70±0.07 / 0.64±0.03 / 89.28±1.77 | 0.74 / 0.62 / 79.97 |
| MMD | 0.79±0.11 / 0.59±0.02 / 61.51±1.79 | 0.71±0.15 / 0.64±0.03 / 91.15±2.33 | 0.79±0.17 / 0.60±0.01 / 86.69±0.19 | 0.76 / 0.61 / 79.87 |
| DANN | 0.70±0.13 / 0.78±0.02 / 47.71±1.56 | 0.79±0.12 / 0.53±0.01 / 84.80±1.14 | 0.77±0.17 / 0.59±0.02 / 58.50±1.74 | 0.75 / 0.64 / 63.67 |
| CDANN | 0.74±0.13 / 0.58±0.02 / 55.87±2.09 | 0.70±0.22 / 0.65±0.02 / 87.06±2.43 | 0.72±0.13 / 0.63±0.02 / 85.76±2.43 | 0.72 / 0.62 / 76.23 |
| DDG | 0.81±0.14 / 0.57±0.03 / 60.08±1.08 | 0.74±0.12 / 0.66±0.03 / 92.53±0.91 | 0.71±0.21 / 0.59±0.03 / **95.02**±1.92 | 0.75 / 0.61 / 82.54 |
| MBDG | 0.79±0.15 / 0.58±0.01 / 60.46±1.90 | 0.73±0.07 / 0.67±0.01 / 94.36±0.23 | 0.71±0.11 / 0.59±0.03 / 93.48±0.65 | 0.74 / 0.61 / 82.77 |
| DDG-FC | 0.76±0.06 / 0.58±0.03 / 59.96±2.36 | 0.83±0.06 / 0.58±0.01 / **96.80**±1.28 | 0.82±0.09 / 0.59±0.01 / 86.38±2.45 | 0.80 / 0.58 / 81.04 |
| MBDG-FC | 0.80±0.13 / 0.58±0.01 / 62.31±0.13 | 0.72±0.09 / 0.63±0.01 / 94.73±2.09 | 0.80±0.07 / 0.53±0.01 / 87.78±2.11 | 0.77 / 0.58 / 81.61 |
| EIIL | **0.87**±0.11 / 0.55±0.02 / 56.74±0.60 | 0.76±0.05 / 0.54±0.03 / 68.99±0.91 | 0.87±0.06 / 0.78±0.03 / 72.19±0.75 | 0.83 / 0.62 / 65.98 |
| FarconVAE | 0.67±0.06 / 0.61±0.03 / 51.21±0.61 | 0.90±0.06 / 0.59±0.01 / 72.40±2.13 | 0.85±0.12 / 0.55±0.01 / 74.20±2.46 | 0.81 / 0.58 / 65.93 |
| FATDM | 0.80±0.10 / 0.55±0.01 / 61.56±0.89 | 0.88±0.08 / 0.56±0.01 / 90.00±0.66 | 0.86±0.10 / 0.60±0.02 / 89.12±1.30 | 0.84 / 0.57 / 80.22 |
| FDDG | **0.87**±0.09 / **0.53**±0.01 / **62.56**±2.25 | **0.94**±0.05 / **0.52**±0.01 / 93.36±1.70 | **0.93**±0.03 / **0.53**±0.02 / 93.43±0.73 | **0.92** / **0.53** / **83.12** |

Table 9: Full performance on `NYSF`. (bold is the best; underline is the second best).

| Methods | DP ↑ / AUC ↓ / Accuracy ↑ | | |
| --- | --- | --- | --- |
| | (R, 0.93) | (B, 0.85) | (M, 0.81) |
| ERM | 0.91±0.07 / 0.53±0.01 / 60.21±1.48 | 0.90±0.07 / 0.54±0.01 / 58.93±1.10 | 0.92±0.04 / 0.54±0.01 / 59.49±1.50 |
| IRM | 0.98±0.01 / 0.52±0.02 / 61.61±0.80 | 0.94±0.04 / **0.52**±0.02 / 56.89±0.73 | 0.92±0.02 / 0.53±0.03 / 59.64±2.33 |
| GDRO | 0.81±0.18 / 0.56±0.02 / 58.73±2.23 | 0.89±0.07 / 0.55±0.03 / 59.44±1.66 | 0.87±0.08 / 0.55±0.02 / **62.57**±0.91 |
| Mixup | 0.96±0.03 / 0.53±0.01 / **62.63**±1.84 | 0.90±0.06 / 0.54±0.04 / 58.96±2.89 | 0.92±0.04 / 0.54±0.03 / 58.29±0.80 |
| MLDG | 0.96±0.03 / 0.52±0.02 / 61.81±0.53 | 0.90±0.08 / 0.55±0.01 / 58.11±0.13 | 0.93±0.02 / 0.53±0.02 / 58.27±0.47 |
| CORAL | 0.95±0.02 / 0.52±0.02 / 62.17±0.92 | 0.93±0.04 / 0.54±0.01 / 58.06±1.99 | 0.95±0.03 / 0.53±0.01 / 58.84±0.74 |
| MMD | 0.91±0.05 / 0.53±0.01 / 60.34±1.39 | 0.89±0.07 / 0.55±0.02 / 58.47±0.35 | 0.92±0.02 / 0.54±0.01 / 59.31±0.40 |
| DANN | 0.83±0.13 / 0.52±0.02 / 40.80±2.47 | **0.96**±0.02 / 0.55±0.03 / 54.55±0.17 | 0.88±0.04 / **0.52**±0.01 / 59.19±1.21 |
| CDANN | 0.95±0.02 / 0.52±0.01 / 57.61±0.68 | 0.94±0.03 / 0.54±0.02 / 56.97±1.29 | 0.87±0.09 / **0.52**±0.02 / 59.59±1.74 |
| DDG | 0.92±0.03 / 0.52±0.01 / 56.52±0.71 | 0.92±0.04 / 0.54±0.04 / 58.21±1.40 | 0.92±0.07 / 0.53±0.02 / 60.91±2.47 |
| MBDG | 0.96±0.02 / 0.52±0.01 / 55.96±1.37 | 0.90±0.07 / 0.70±0.01 / 51.52±1.55 | **0.96**±0.02 / 0.53±0.03 / 58.74±2.46 |
| DDG-FC | 0.95±0.03 / 0.52±0.01 / 54.53±1.44 | 0.93±0.02 / 0.53±0.03 / 59.32±0.59 | 0.92±0.04 / 0.52±0.01 / 60.08±1.31 |
| MBDG-FC | 0.96±0.02 / 0.55±0.02 / 55.93±1.98 | 0.91±0.07 / 0.54±0.03 / 55.50±0.55 | 0.90±0.06 / 0.53±0.02 / 57.37±2.39 |
| EIIL | 0.95±0.02 / 0.52±0.02 / 58.28±3.23 | 0.92±0.03 / 0.54±0.02 / 56.76±3.87 | 0.83±0.11 / 0.54±0.02 / 59.47±1.69 |
| FarconVAE | 0.90±0.07 / 0.53±0.03 / 60.52±0.14 | 0.89±0.05 / 0.55±0.04 / **60.30**±0.64 | 0.82±0.07 / 0.56±0.01 / 60.31±0.40 |
| FATDM | 0.93±0.05 / 0.52±0.01 / 59.32±1.00 | 0.86±0.05 / 0.58±0.02 / 59.01±0.32 | 0.85±0.08 / 0.53±0.02 / 60.45±0.87 |
| FDDG | **0.99**±0.00 / **0.50**±0.00 / 62.01±1.87 | **0.96**±0.01 / **0.52**±0.02 / 58.37±0.67 | 0.92±0.02 / **0.52**±0.02 / 59.49±1.93 |

| Methods | DP ↑ / AUC ↓ / Accuracy ↑ | | |
| --- | --- | --- | --- |
| | (Q, 0.59) | (S, 0.62) | **Avg** |
| ERM | 0.88±0.06 / 0.57±0.02 / 62.48±0.64 | 0.86±0.12 / 0.61±0.03 / 54.54±0.68 | 0.90 / 0.56 / 59.13 |
| IRM | 0.87±0.06 / 0.54±0.01 / 55.81±1.74 | 0.89±0.07 / 0.54±0.03 / 57.00±2.01 | 0.92 / 0.53 / 58.19 |
| GDRO | 0.86±0.05 / 0.57±0.01 / 62.92±1.17 | 0.77±0.08 / 0.64±0.04 / 60.44±2.86 | 0.84 / 0.57 / **60.82** |
| Mixup | 0.93±0.04 / 0.53±0.01 / 61.34±1.60 | 0.84±0.08 / 0.61±0.02 / 53.07±3.13 | 0.91 / 0.55 / 58.86 |
| MLDG | 0.89±0.08 / 0.56±0.02 / 62.85±2.38 | 0.85±0.05 / 0.59±0.03 / 54.42±0.02 | 0.91 / 0.55 / 59.10 |
| CORAL | 0.95±0.03 / 0.53±0.02 / 61.45±0.28 | 0.88±0.08 / 0.54±0.03 / 52.08±1.06 | 0.93 / 0.53 / 58.52 |
| MMD | 0.88±0.03 / 0.56±0.01 / 62.48±1.31 | 0.81±0.17 / 0.61±0.02 / 57.73±1.54 | 0.88 / 0.56 / 59.67 |
| DANN | 0.96±0.02 / 0.53±0.02 / 63.60±0.34 | 0.86±0.05 / 0.56±0.03 / 58.96±0.98 | 0.90 / 0.54 / 55.42 |
| CDANN | 0.97±0.02 / 0.54±0.03 / **64.25**±1.25 | 0.74±0.16 / 0.60±0.01 / 57.73±1.89 | 0.89 / 0.54 / 59.23 |
| DDG | 0.89±0.07 / 0.55±0.01 / 56.68±0.87 | 0.84±0.07 / 0.58±0.03 / 54.91±1.33 | 0.90 / 0.54 / 57.44 |
| MBDG | 0.96±0.03 / 0.52±0.01 / 60.73±1.56 | 0.90±0.04 / **0.52**±0.02 / 52.45±1.98 | 0.93 / 0.56 / 55.88 |
| DDG-FC | 0.92±0.02 / 0.54±0.02 / 59.90±1.75 | 0.90±0.05 / 0.57±0.02 / 57.45±0.08 | 0.92 / 0.53 / 58.26 |
| MBDG-FC | 0.94±0.04 / 0.52±0.01 / 61.04±2.31 | 0.91±0.06 / 0.53±0.03 / 52.57±0.92 | 0.92 / 0.53 / 56.48 |
| EIIL | 0.84±0.12 / 0.55±0.02 / 52.18±0.26 | 0.95±0.03 / 0.59±0.02 / 55.74±0.12 | 0.90 / 0.54 / 56.49 |
| FarconVAE | 0.97±0.02 / 0.56±0.03 / 61.30±1.14 | 0.86±0.10 / 0.58±0.02 / 60.70±1.48 | 0.89 / 0.56 / 60.62 |
| FATDM | 0.85±0.05 / 0.52±0.01 / 60.35±0.44 | 0.88±0.03 / **0.52**±0.01 / 59.22±0.09 | 0.87 / 0.53 / 59.67 |
| FDDG | **0.99**±0.01 / **0.50**±0.00 / 59.11±0.94 | **0.98**±0.02 / 0.53±0.01 / **60.77**±0.23 | **0.97** / **0.51** / 59.95 |

Table 10: Ablation studies results on `ccMNIST`.

| Methods | DP ↑ / AUC ↓ / Accuracy ↑ | | | |
| --- | --- | --- | --- | --- |
| | (R, 0.11) | (G, 0.43) | (B, 0.87) | **Avg** |
| FDDG w/o sf | 0.23±0.05 / 0.98±0.01 / 94.89±1.72 | 0.11±0.06 / 0.92±0.02 / 98.19±1.39 | 0.42±0.06 / 0.72±0.03 / 95.28±0.22 | 0.25 / 0.87 / 96.12 |
| FDDG w/o $T$ | 0.21±0.12 / 0.92±0.01 / 96.74±1.15 | 0.15±0.08 / 0.86±0.02 / 96.95±0.93 | 0.48±0.06 / 0.57±0.02 / 96.05±1.17 | 0.28 / 0.79 / 96.58 |
| FDDG w/o fc | 0.22±0.08 / 0.91±0.02 / 96.63±0.63 | 0.44±0.16 / 0.75±0.01 / 97.90±0.40 | 0.97±0.02 / 0.61±0.02 / 96.01±0.20 | 0.54 / 0.76 / 96.85 |

Table 11: Ablation studies results on `FairFace`.

| Methods | DP ↑ / AUC ↓ / Accuracy ↑ | | |
|---|---|---|---|
| | (B, 0.91) | (E, 0.87) | (I, 0.58) |
| FDDG w/o sf | 0.68±0.18 / 0.57±0.02 / 93.07±0.68 | 0.43±0.20 / 0.60±0.03 / 95.55±2.09 | 0.37±0.09 / 0.59±0.03 / 92.26±0.37 |
| FDDG w/o $T$ | 0.83±0.08 / 0.56±0.01 / 92.81±0.81 | 0.50±0.22 / 0.56±0.01 / 95.12±0.73 | 0.42±0.17 / 0.59±0.02 / 92.34±0.14 |
| FDDG w/o fc | 0.59±0.16 / 0.58±0.01 / 92.92±1.35 | 0.36±0.08 / 0.62±0.03 / 95.55±1.84 | 0.42±0.20 / 0.62±0.02 / 93.35±0.83 |

| Methods | DP ↑ / AUC ↓ / Accuracy ↑ | | |
|---|---|---|---|
| | (M, 0.87) | (S, 0.39) | (W, 0.49) |
| FDDG w/o sf | 0.49±0.13 / 0.62±0.03 / 92.61±2.32 | 0.69±0.22 / 0.56±0.01 / 93.28±2.31 | 0.35±0.26 / 0.58±0.01 / 92.18±0.46 |
| FDDG w/o $T$ | 0.39±0.07 / 0.68±0.01 / 91.46±2.05 | 0.92±0.06 / 0.56±0.01 / 87.87±1.25 | 0.52±0.23 / 0.59±0.01 / 90.78±0.31 |
| FDDG w/o fc | 0.38±0.15 / 0.72±0.03 / 92.27±0.02 | 0.42±0.16 / 0.67±0.03 / 92.17±0.99 | 0.34±0.08 / 0.72±0.03 / 91.88±0.67 |

| Methods | DP ↑ / AUC ↓ / Accuracy ↑ | |
|---|---|---|
| | (L, 0.48) | Avg |
| FDDG w/o sf | 0.47±0.07 / 0.63±0.01 / 92.62±0.93 | 0.49 / 0.59 / 93.08 |
| FDDG w/o $T$ | 0.53±0.03 / 0.59±0.01 / 91.19±0.57 | 0.58 / 0.59 / 91.65 |
| FDDG w/o fc | 0.40±0.07 / 0.70±0.02 / 92.96±0.85 | 0.42 / 0.66 / 93.01 |

Table 12: Ablation studies results on `YFCC100M-FDG`.

| Methods | DP ↑ / AUC ↓ / Accuracy ↑ | | | |
|---|---|---|---|---|
| | $(d_0, 0.73)$ | $(d_1, 0.84)$ | $(d_2, 0.72)$ | Avg |
| FDDG w/o sf | 0.69±0.13 / 0.57±0.02 / 43.09±1.45 | 0.83±0.08 / 0.63±0.02 / 89.68±0.60 | 0.89±0.05 / 0.54±0.03 / 87.70±1.69 | 0.80 / 0.58 / 73.49 |
| FDDG w/o $T$ | 0.82±0.12 / 0.56±0.03 / 47.21±1.17 | 0.83±0.05 / 0.63±0.01 / 73.10±0.26 | 0.82±0.08 / 0.53±0.02 / 72.95±2.25 | 0.82 / 0.57 / 64.42 |
| FDDG w/o fc | 0.72±0.17 / 0.69±0.03 / 54.24±1.75 | 0.92±0.02 / 0.64±0.03 / 94.35±2.35 | 0.92±0.07 / 0.64±0.03 / 93.20±2.17 | 0.86 / 0.66 / 80.59 |

Table 13: Ablation studies results on `NYSF`.

| Methods | DP ↑ / AUC ↓ / Accuracy ↑ | | |
|---|---|---|---|
| | (R, 0.93) | (B, 0.85) | (M, 0.81) |
| FDDG w/o sf | 0.95±0.02 / 0.52±0.01 / 55.78±1.01 | 0.97±0.01 / 0.51±0.01 / 55.30±1.08 | 0.95±0.03 / 0.53±0.01 / 58.29±0.80 |
| FDDG w/o $T$ | 0.95±0.03 / 0.52±0.01 / 61.36±0.42 | 0.91±0.06 / 0.54±0.01 / 57.67±0.82 | 0.89±0.05 / 0.55±0.01 / 60.68±0.31 |
| FDDG w/o fc | 0.95±0.02 / 0.52±0.02 / 63.72±0.37 | 0.87±0.09 / 0.55±0.01 / 58.86±0.68 | 0.89±0.08 / 0.54±0.01 / 60.61±0.59 |

| Methods | DP ↑ / AUC ↓ / Accuracy ↑ | | |
|---|---|---|---|
| | (Q, 0.59) | (S, 0.62) | Avg |
| FDDG w/o sf | 0.92±0.06 / 0.54±0.02 / 57.61±1.30 | 0.90±0.02 / 0.59±0.02 / 52.82±1.20 | 0.94 / 0.53 / 55.96 |
| FDDG w/o $T$ | 0.97±0.02 / 0.52±0.01 / 59.33±0.17 | 0.87±0.11 / 0.57±0.01 / 55.40±0.73 | 0.92 / 0.54 / 58.89 |
| FDDG w/o fc | 0.83±0.08 / 0.57±0.01 / 64.17±0.35 | 0.89±0.06 / 0.58±0.02 / 56.51±0.84 | 0.89 / 0.55 / 60.77 |

Table 14: Trade-off between fairness-accuracy.

| | DP ↑ / AUC ↓ / Accuracy ↑ | | | |
|---|---|---|---|---|
| | ccMNIST | FairFace | YFCC100M-FDG | NYSF |
| $\lambda_2 = 0.05$ | 0.53 / 0.75 / 98.61 | 0.57 / 0.63 / 95.99 | 0.88 / 0.55 / 88.31 | 0.86 / 0.57 / 61.71 |
| $\lambda_2 = 1$ | 0.71 / 0.66 / 96.76 | 0.70 / 0.58 / 93.42 | 0.92 / 0.53 / 83.12 | 0.97 / 0.51 / 59.95 |
| $\lambda_2 = 50$ | 0.72 / 0.63 / 89.07 | 0.78 / 0.52 / 88.65 | 0.95 / 0.51 / 72.63 | 0.98 / 0.51 / 55.28 |
| $\gamma_1 = 0.025, \gamma_2 = 0.25$ | 0.47 / 0.79 / 97.07 | 0.53 / 0.60 / 93.99 | 0.88 / 0.55 / 88.69 | 0.86 / 0.56 / 61.71 |
| $\gamma_1 = 0.25, \gamma_2 = 0.025$ | 0.66 / 0.75 / 88.54 | 0.62 / 0.58 / 93.06 | 0.91 / 0.54 / 81.49 | 0.86 / 0.57 / 58.03 |
| $\gamma_1 = 0.025, \gamma_2 = 0.025$ | 0.71 / 0.66 / 96.97 | 0.70 / 0.58 / 93.42 | 0.92 / 0.53 / 83.12 | 0.97 / 0.51 / 59.95 |

# G  LIMITATIONS

In Sec. 5 and Appendix F, we empirically demonstrate the effectiveness of the proposed FDDG, wherein our method is developed based on assumptions. We assume (1) data instances can be encoded into three latent factors, (2) such factors are independent of each other, and (3) each domain shares the same content space. FDDG may not work well when data are generated with more than three factors, and each is correlated to the other. To address such limitations, studies on causal learning could be a solution. Moreover, our model relies on domain augmentation. While the results demonstrate its effectiveness, it might not perform optimally when content spaces do not completely overlap across domains. In such scenarios, a preferable approach would involve initially augmenting data by minimizing semantic gaps for each class across training domains, followed by conducting domain augmentations.