# OpenReview forum: "Fairness-Aware Domain Generalization under Covariate and Dependence Shifts"
_ICLR.cc/2024/Conference — Submitted to ICLR 2024_

### Official Review · Reviewer_FJko · 2023-10-25

**Soundness:** 3 good
**Presentation:** 1 poor
**Contribution:** 2 fair
**Rating:** 5
**Confidence:** 3

**Summary:**

The authors introduce a novel approach to multi-source domain generalization, addressing not only predictive performance but also fairness guarantees. They account for distribution shifts arising from covariate shifts and variations in the relationship between the sensitive attribute and the class label (spurious correlation). Their key assumption is that all source instances result from a transformation and that instances from one domain can be transformed into instances of another. Their framework begins by training a transformation model, followed by the generation of synthetic instances from previously unseen potential target domains. The classifier is subsequently trained on a combination of synthetic and real instances.

**Strengths:**

S1 - The issue being investigated is of significant importance. Domain generalization, considering both covariate shift and spurious correlations, is highly relevant to real-world scenarios where such challenges frequently manifest. Their proposed approach appears to be well-suited for addressing situations commonly encountered in practical applications and can be applied under realistic conditions.

S2 - The introduction effectively substantiates all the claims made, including the contributions put forth by the authors. These assertions find validation through a thorough description of the methodology employed and the experiments conducted. The method section elaborates on the techniques and approaches considered, demonstrating how they align with the stated objectives. Furthermore, the experimental results provide empirical evidence that supports the claims made in the introduction.

S3 - The authors have invested substantial effort to establish a rigorous and thorough evaluation process within the paper. They have taken into account an extensive set of baseline models and addressed three distinct classification tasks. Additionally, the authors have conducted meticulous ablation studies, scrutinizing various factors to guarantee a robust and comprehensive evaluation.

S4 - The authors offer robust mathematical solutions to enable the training of their approach.

S5 - The paper demonstrates commendable attention to reproducibility by providing thorough and detailed information regarding the experimental setup. Furthermore, the ablation studies provide insightful analysis.

S6 - The paper is well structured with well-defined sections and subsections, contributing to its overall readability.  The paper includes sub-sections that effectively delineate different aspects of the methodology, ensuring a coherent and structured presentation.

**Weaknesses:**

W1 - The paper would benefit from enhanced clarity in its writing. At times, the authors employ technical language without providing accompanying explanations, particularly in the introduction. For instance, a more thorough contextualization of spurious correlations, why they manifest in neural networks, and when they become problematic would greatly assist readers. Additionally, the paper references concepts such as the latent content space, sensitive factor, and style factor without offering any accompanying explanations or clarifications.
W2 - Furthermore, there are certain statements that are not entirely accurate and can be misleading. For instance, the statement 'existing methods assume that the mentioned distribution shifts remain static, and there exist distinct fairness dependence between model outcomes' raises questions about the term 'fairness dependence.' Fairness guarantees are typically associated with model outcomes, so the variable component is not fairness itself, but rather the dependence between the sensitive or spurious feature and the class label. Similarly, the sentence 'the fairness patterns between domains remain constant' should ideally clarify that it is the patterns of dependence between domains that remain constant, not necessarily fairness patterns.

W3 - The authors assert multiple times that they are introducing a novel form of shift. However, it's important to note that there is an extensive body of literature that has previously examined the effects of spurious correlations when the relationship between the spurious feature and the class label varies across domains. Furthermore, in the field of fairness, there has been prior research on shifts within subpopulations. Hence, the extent to which this claim holds true remains somewhat uncertain.


W4 - On page 5, the statement 'Enforcing constraints on deep networks is known to be a challenging problem' is made without prior reference to any specific type of model. It appears as if the authors assume that the reader implicitly understands that this approach pertains to neural networks, even though they do not explicitly mention specific models anywhere in the text. To address this, the authors could, in the introduction, specify that the presence of spurious correlations is a common issue in deep neural networks (DNNs), thereby clarifying that they are dealing with a particular family of functions. Throughout the text, especially when explaining their approach, it would be beneficial for them to explicitly mention that they are considering neural networks. The absence of such explicit references may indeed present a clear issue.

W5 - The authors initially discuss the challenge of maintaining fairness guarantees under distribution shift, particularly when there are varying dependencies between the sensitive feature and the class feature (a particular case of spurious correlation, when the spurious feature is the sensitive attribute). However, the example they present in the introduction to motivate the problem (Figure 2) pertains to the broader issue of spurious correlations rather than being specific to fairness concerns. In this context, they consider the background as the spurious feature, not necessarily linked to any sensitive attribute. Moreover, in the experimental section, the spurious feature is not always synonymous with a sensitive attribute. As a result, the paper's objective may not be entirely clear, as it's uncertain whether it primarily addresses fairness issues in multi-source domain generalization or aims for a more general approach dealing with subgroup robustness in domain generalization when faced with varying spurious correlations across domains.

W6 - I have noticed that the information provided about fairness concepts in the paper is rather limited. In the paragraph discussing fairness notions, the authors do not acknowledge the existence of various definitions for quantifying the fairness guarantees of a classifier, including statistical notions, individual notions, and mini-max notions. They focus exclusively on statistical notions and, within this category, specifically on demographic parity (DP).
It is essential to note that the definition presented in the paper about statistical notions pertains solely to DP. The broader definition of a statistical notion of fairness encompasses parity in a given performance measure across different groups, with the specific terminology of these statistical notions varying based on the performance metric under consideration. Therefore, the paper's definition is not entirely accurate and should be revised.
Furthermore, it is advisable for the authors to reference relevant literature that offers an extensive review of existing fairness metrics, providing readers with a more comprehensive understanding of the available fairness measures.

W7 - Also in connection with fairness notions, there are statements that could be potentially misleading. For instance, the assertion that 'the fairness notion $\rho(\hat{Y}, Z)$ is defined as the difference of demographic parity (DP)' might not be accurate. DP itself pertains to the difference in predictions, and, as such, fairness concerns relate to disparities in acceptance rates rather than the difference in DP.

W8 - In the experimental section, the authors evaluate various domain generalization methods, including GDRO, Mixup, and CORAL, which are originally designed for single-source domain scenarios. This may result in an unfair comparison, as they are applied to multi-source data even though they were not explicitly designed for such cases. Moreover, the authors do not clarify that certain baseline domain generalization methods are intended for either single-source or multi-source domain settings, which is a crucial distinction to highlight for a fair evaluation. Additionally, the paper does not provide clear information on how they adapt single-domain methods to handle multiple domain data.

W9 - In the section addressing related works, specifically within the paragraph titled 'Fairness learning for changing environments,' the references provided are notably limited. Notably, several domain adaptation works in fairness have been explored, but these are not mentioned in the current text. It would be beneficial to refer to [1], where readers can access a comprehensive overview of the field about approaches aimed at maintaining fairness guarantees of classifiers in evolving or changing environments. Additionally, in the context of domain generalization amidst spurious correlations, it's worth noting that the paper lacks references to more recent works, such as [2].


W10 - The paper contains several issues related to its references. Firstly, in the introduction, there is a duplication of references, specifically for Pham et al. and Robey et al. Additionally, there is a need to differentiate between explicit and implicit citations. For instance, in the sentence 'in addition to the two distribution shifts stated in (Robey et al., 2021),' the reference should appear without parentheses. Moreover, within the list of references, there are arXiv references for papers that have already been published, such as in the case of (Sagawa et al.) which was published as a conference paper at ICLR 2020.

W11 - No constraints or limitations explicitly pointed out in the main text.

[1] Barrainkua, A., Gordaliza, P., Lozano, J. A., & Quadrianto, N. (2022). A Survey on Preserving Fairness Guarantees in Changing Environments. arXiv preprint arXiv:2211.07530.

[2] Kirichenko, P., Izmailov, P., & Wilson, A. G. (2022, September). Last Layer Re-Training is Sufficient for Robustness to Spurious Correlations. In The Eleventh International Conference on Learning Representations.

**Questions:**

Q1 - What was the rationale behind selecting the concept of demographic parity (DP)? Why was the consideration of worst-group accuracy, a conventional fairness indicator in the literature concerning generalization challenges amid spurious correlations, not included in your study?
Q2 - Is it feasible to readily extend this approach to accommodate other fairness concepts, such as Equality of Opportunity, which requires access to true labels? Can it be adapted to address mini-max notions of fairness?
Q3 - How do you adapt single-domain methods to handle multiple domain data?
Q4 - Is the existence of T always guaranteed? Are there any limitations w.r.t.  T?
Q5 - Can this be applied in multi-dimensional sensitive attributes, or in the presence of intersectional groups?
Q6 - Does your model have constraints or limitations?
Q7 - How does the computational burden of your method compare to state-of-the-art approaches?

---

> ### Author Response · Authors · 2023-11-22
>
> We sincerely appreciate the reviewer's valuable and insightful suggestions, which have significantly enhanced the clarity of our writing. We are committed to carefully considering all recommendations to further refine and improve the quality of our paper.
>
> [W3] Some domain generalization papers concern the shift in fairness, such as [a, b]. For example, in [a], they gave an example using the Waterbird dataset, where the backgrounds (i.e., sensitive subgroups, land and water) are spuriously correlated with class labels (waterbirds and landbirds). This spurious correlation causes sub-par performance in the smallest subgroups. However, [a] only takes the shift in fairness as domain variations. In our paper, we state the distribution shift across domains due to covariate shifts (data features) and changes in fair dependence. Therefore, the dependence shift is proposed. Since prior research, such as [a, b], has been proposed, we do not conclude the dependence shift is a major contribution to our paper.
>
> [W5] In our setting, an image has two labels, a class label and a sensitive label. For example, an image of a “dog in grass” has a “dog” class label and a “grass” sensitive label. Therefore, the features extracted from this image can be perceived as encapsulating information about both the "dog" and "grass". Another example is given in the empirical study using the ccMNIST dataset where the digit color is used as domain variation due to covariate shift and the background color is used to describe the sensitive label, dark and black. This is similar to traditional fairness learning using tabular datasets. For example, gender is a sensitive label of instance, “height” and “has a beard” could be features used to describe gender information.
>
> [W6, Q2, Q5] This paper focuses on group fairness and uses DP or DDP (difference of DP) as the fairness notion. However, in our Eq.(1), it can be easily changed to the difference of equalized odds when setting $p_1=\mathbb{P}(Z=1, Y=1)$. And Eq.(1) can be generalized to take the maximum of $g$ across different classes for the multi-class setting. For multi-dimensional sensitive attributes, Eq.(1) can be replaced by corresponding group fairness metrics, such as an extended version of DP or EO proposed in [e].
>
> [W8, Q3] GDRO, Mixup, and CORAL are used as baseline methods in many multi-source domain generalization works, such as [c] and [d]. Such methods have been included in a public code repository developed by FacebookResearch, named domainBed (see https://github.com/facebookresearch/DomainBed) with 1.2k stared. Moreover, in our Problem 1, the max operator requires domain labels $e$. However, the domain labels are expensive or even impossible to obtain in part due to privacy issues. Therefore, under the disentanglement-based invariance and domain shift assumptions, Problem 1 can be approximated to Problme 2 with the max operator eliminated.
>
> [W9, W10] Thanks for the head-up. Certainly, addressing the related work as highlighted by the reviewer and making the necessary corrections regarding citations will be a part of our discussion and subsequent revisions.
>
> [W11, Q4, Q6, Q7] Some limitations could be the assumptions that we made (1) instances can be encoded into three latent factors (2) such factors are independent of each other, and (3) each training domain shares the same content space. Our proposed method may not work well when data are generated with more than three factors, and each is correlated to the other. Besides, our model is built based on domain augmentation. Although results showcase the effectiveness of our model, it may not work well when content spaces are not fully overlapped across domains. In this case, it would be better to first augment data by minimizing semantic gaps for each class across training domains and then conduct domain augmentations. To address such limitations, studies on causal learning could be a solution, and we will save it for our future works. Besides, our model is a bit expensive as the transformation model is learned using GAN. The detailed training process is discussed in the Appendix.
>
> [a] Creager et al., Environment Inference for Invariant Learning, ICML 2021
>
> [b] Oh et al., Learning Fair Representation via Distributional Contrastive Disentanglement, KDD 2022
>
> [c] Zhang et al., Towards principled disentanglement for domain generalization. CVPR 2022
>
> [d] Robey et al.,  Model-based domain generalization. NeurIPS 2021
>
> [e] Yang et al., Fairness with Overlapping Groups, NeurIPS 2020

---

### Official Review · Reviewer_55mH · 2023-10-31

**Soundness:** 2 fair
**Presentation:** 2 fair
**Contribution:** 3 good
**Rating:** 5
**Confidence:** 4

**Summary:**

This paper introduces a machine learning model designed to deliver fair predictions for an unseen target domain in a domain generalization scenario. In particular, the authors first introduce a novel concept called "dependence shift" to capture the relationship between labels and sensitive attributes. Then, they propose a two-stage approach to achieve demographic parity on the target domain while dealing with covariate and dependence shifts. In the first stage, the authors develop a transformation model to address covariate shift assumptions. In the second stage, they train the model with fairness regularizations to ensure fair predictions on the target domain in the presence of dependence shifts. Experimental results on several datasets demonstrate that the proposed method attains a more favorable trade-off between fairness and accuracy on target domains compared to baseline approaches.

**Strengths:**

- This paper addresses fairness in the context of distribution shifts which is an important topic in fair machine learning.

- The availability of code and supplementary documentation enhances the reproducibility of the research.

**Weaknesses:**

- The motivation behind "dependence shift" is not entirely clear. The example presented in Figure 1 is somewhat ambiguous. It's unclear why attributes like "grass" and "couch" are considered sensitive. The authors should consider providing a more realistic example to better illustrate their problem setting.

- The authors mention the potential extension of their work to handle settings with multi-class, multi-sensitive attributes, and other fairness concepts. However, it is not straightforward how their results can be extended to these scenarios. More details are needed to clarify this aspect.

- There's an existing work that also deals with fairness in domain generalization [cite]. That work doesn't assume invariance of $\rho$ across domains which then imply that it can also handle the dependence shift. Thus, it's essential for the authors to differentiate their work from this previous research. Additionally, Theorem 1 appears quite similar to Theorem 3 in [cite]. The authors should highlight the novel insights offered by Theorem 1 compared to the existing result.

- Assumption 1 appears more like a definition than an assumption since a mapping between two domains can always be constructed. For instance, when domains share the same support, an identity mapping fulfills this assumption.

- The theoretical results lack coherency. It is not entirely clear how Theorems 1 and 2 relate to the proposed algorithm's goals, which involve learning a transformation model to address covariate shift and constructing a fairness-aware T-invariance classifier that achieves demographic parity.

- The authors should explain why they restrict the hypothesis class $\mathcal{F}$ to the set of invariant fairness-aware classifiers and why the max operator in the objective function can be eliminated in Problem 2.

- To strengthen their claims, the authors should consider including comparisons of fairness-accuracy trade-off curves (Pareto frontiers) for the models in the experimental section. This would provide stronger evidence for their proposed method's effectiveness.

**Questions:**

Please see the Weaknesses.

---

> ### Author Response · Authors · 2023-11-22
>
> [W1] Our example in Figure 1 is motivated by [a]. In [a], they gave an example using the Waterbird dataset, where the backgrounds (i.e., sensitive subgroups, land, and water) are spuriously correlated with class labels (waterbirds and landbirds). This spurious correlation causes sub-par performance in the smallest subgroups. Similar to Figure 2 in our paper, the spurious correlation between classes (dogs and cats) between image backgrounds or other objects (grass and couches) leads to untrusted predictions. For example, if a model learned from all training images “dogs in the grass” and “cats in the couch”, as a consequence the model face challenges accurately classifying “dogs” and “cats” during inference with images “dogs in the couch” and “cats in the grass”. This indicates sensitive features “grass” and “couch” are implicitly correlated with model prediction.
>
> [W2]  In this paper, we use a binary sensitive attribute and classes just as a simple case for idea illustration, as many fairness works did, such as [a]. This can be easily extended to multinary cases by either changing the group fairness notion or taking the maximum of difference of DP over each class/subgroup. Similar settings are considered in [b].
>
> [W3] The reviewer seems to forget to give the [cite] reference, but we assume it refers to [b]. The difference between [b] and our works, with respect to handling fairness (dependence shift), is that [b] does not implicitly or explicitly assume fairness levels (dependence between sensitive attributes and classes) change across domains. Since test domains are inaccessible during training, in our works training domains are augmented with new variations and fairness levels. The greater the diversity of domains encompassing various variations and fairness levels encountered during training, the more effectively the model can generalize to different domains during inference. With limited fairness levels in training domains, in contrast to [b], our model has a better capability of generalization in fairness.
>
> [W4] Some other papers in domain generalization based on maximizing the Evidence Lower Bound (ELBO) do not make this assumption, such as [c]. We hence consider it as an assumption but not a definition.
>
> [W5] Theorem 1 provides an upper bound on the fairness adaptation achievable using the proposed algorithm. Theorem 2 gives the empirical duality gap, as the proposed algorithm using duality that iteratively updates model primal parameters and dual variables (lines 11-12).
>
> [W6] The goal of domain generalization problems is to find an invariant classifier in the classifier space of $\mathcal{F}$, theoretically. But in experiments, with given datasets, this goal is required to approximate to find model parameters of the classifier in a parametric space. Moreover, in Problem 1, the max operator requires domain labels $e$. However, the domain labels are expensive or even impossible to obtain in part due to privacy issues. Therefore, under the disentanglement-based invariance and domain shift assumptions, Problem 1 can be approximated to Problme 2 with the max operator eliminated. We will clearly explain this in an updated version.
>
>
>
> [a] Creager et al., Environment Inference for Invariant Learning, ICML 2021
>
> [b] Pham et al., Fairness and accuracy under domain generalization. ICLR 2023
>
> [c] Liu et al., Learning Causal Semantic Representation for Out-of-Distribution Prediction, NeurIPS 2021

---

> ### Author Response · Authors · 2023-11-22
>
> [W7] Some additional results on the trade-off between fairness-accuracy are conducted. We will update it in a later version.
>
> In our algorithm, because $\lambda_2$  is the parameter that regularizes the fair loss, we conduct additional experiments to show the change between accuracy and fairness. Our results show that the larger (smaller) $\lambda_2$ , the better (worse) model fairness for each domain as well as in average, but it gives worse (better) model utility.
> | |      DP $\uparrow$ / AUC $\downarrow$ / Accuracy $\uparrow$|  |  |  |
> |----------|:-------------:|:------:|:------:|:------:|
> |                    	| ccMNIST                                                	| FairFace            	| YFCC                	| NYSF                	|
> | $\lambda_2 = 0.05$ | 0.53 / 0.75 / 98.61 | 0.57 / 0.63 / 95.99 | 0.88 / 0.55 / 88.31 | 0.86 / 0.57 / 61.71 |
> | $\lambda_2 = 1$ (results in the paper) |    0.71 / 0.66 / 96.76 | 0.70 / 0.58 / 93.42 | 0.92 / 0.53 / 83.12 | 0.97 / 0.51 / 59.95 |
> | $\lambda_2 = 50$ | 0.72 / 0.63 / 89.07 | 0.78 / 0.52 / 88.65 | 0.95 / 0.51 / 72.63 | 0.98 / 0.51 / 55.28 |
>
> Moreover, we show additional experiment results based on choosing different  $\gamma_1$ and $\gamma_2$. We observe that (1) by only increasing $\gamma_2$ , the model towards giving unfair outcomes but higher accuracy; (2) by only increasing $\gamma_1$, performance on both model fairness and accuracy decreases. This may be due to the failure of disentanglement of factors.
>
> | |      DP $\uparrow$ / AUC $\downarrow$ / Accuracy $\uparrow$|  |  |  |
> |----------|:-------------:|:------:|:------:|:------:|
> |                    	| ccMNIST                                                	| FairFace            	| YFCC                	| NYSF                	|
> | $\gamma_1= 0.025, \gamma_2=0.25$ | 0.47 / 0.79 / 97.07 | 0.53 / 0.60 / 93.99 | 0.88 / 0.55 / 88.69 | 0.86 / 0.56 / 61.71 |
> | $\gamma_1= 0.25, \gamma_2=0.025$ |    0.66 / 0.75 / 88.54 | 0.62 / 0.58 / 93.06 | 0.91 / 0.54 / 81.49 | 0.86 / 0.57 / 58.03 |
> | $\gamma_1= 0.025, \gamma_2=0.025$ (results in the paper) | 0.71 / 0.66 / 96.97 | 0.70 / 0.58 / 93.42 | 0.92 / 0.53 / 83.12 | 0.97 / 0.51 / 59.95 |

---

### Official Review · Reviewer_hRwe · 2023-11-03

**Soundness:** 2 fair
**Presentation:** 1 poor
**Contribution:** 2 fair
**Rating:** 3
**Confidence:** 4

**Summary:**

This paper addresses the problem of domain generalization and fairness simultaneously under covariate and sensitive dependence shifts. The authors introduce dependence shift, which implies the dependence of the sensitive attribute and the outcome variable varies across domains. The authors formulate the problem as a minimax optimization problem with demographic parity constraint. Experiments were done on several benchmark datasets.

**Strengths:**

- This paper addresses a domain generalization with the consideration of fairness, which is an important but less explored subject in machine learning and related fields.

**Weaknesses:**

- I do not think the examples given in Figure 1 are appropriate: (1) concept shift is the disparity in $Y^e | X^e$. It doesn't mean a completely new label appears in the target domain; (2) I am unsure if such an auxiliary background or objective, such as couch or grass, is an appropriate "sensitive attribute." I think those can be spurious features; however, often, in real-world cases, sensitive attributes are more than just spurious features as they correlate with both input and output. Moreover, ensuring fairness across the images that include a couch or grass does not seem to be a convincing example of fairness.
- Also, in Figure 2, I do not think what the authors presented is "group fairness levels," as the fairness notion considered in the study is demographic parity, which is calculated as a statistical dependence of the prediction and the sensitive attributes. However, in the figure, no prediction is considered. I think what the authors presented there is the dependence (or correlation) between the true label and the sensitive attribute.
- In many applications, the sensitive attribute and outcome variable are nonbinary. The extension to such cases of the approach introduced in the study does not seem to be straightforward.
- I found the evaluation metrics are inconsistent - why do the authors evaluate the fairness in difference of AUC but the performance in accuracy?
- The authors used so-called inter-group AUC difference as a fairness metric. There have been extensive discussions on different versions of AUC differences, and it has been demonstrated that inter-group AUC difference does not reflect all possible performance disparities. Please refer to [Kallus and Zhou (2019)](https://papers.nips.cc/paper_files/paper/2019/hash/73e0f7487b8e5297182c5a711d20bf26-Abstract.html), [Yang et al. (2023)](https://dl.acm.org/doi/abs/10.1609/aaai.v37i10.26405) and the references therein.
- It seems like the authors confuse demographic disparity (dependence of $\hat{Y}$ and $Z$) and dependence of $Y$ and $Z$. E.g., in Figure 2 and Table 1, the authors presented the degrees of dependence of $Y$ and $Z$ as if they were fairness measurements.
- Some important related works are missing. E.g., [Rezaei et al. (2021)](https://ojs.aaai.org/index.php/AAAI/article/view/17135), [Singh et al. (2021)](https://dl.acm.org/doi/10.1145/3442188.3445865), [Giguere et al. (2022)](https://openreview.net/forum?id=wbPObLm6ueA), [Chen et al. (2022)](https://openreview.net/forum?id=U3gobB4oKv). This is not an exhaustive list.

**Questions:**

- Could the authors prove or give some thoughts about the existence of a non-trivial solution to Problem 1?
- Is there any reason the authors used DP as the fairness criterion? Many previous studies revealed that the DP might not be an ideal fairness criterion.
- The results show that the proposed method effectively reduces the AUC difference across sensitive groups in almost all cases. Could the authors kindly explain why this happened? Because, for me, the relationship between the statistical dependence of the model prediction and sensitive attribute and the AUC differences is unclear. Thus, reducing the dependence (or equivalently achieving demographic parity) does not necessarily result in smaller AUC differences. To the best of my knowledge, a similar analysis has been done by [Zhao and Gordan (2022)](https://dl.acm.org/doi/abs/10.5555/3586589.3586646). They proved that fair representation (that is, independent of the sensitive attribute, which satisfies SP) results in accuracy parity (the same accuracies across all sensitive groups) if the Bayes optimal classifiers across different groups are close.

---

> ### Author Response · Authors · 2023-11-22
>
> [W1] (1) Concept shift in Figure1 is more related to semantic shift. Semantic shift as illustrated in [a] can be viewed as a type of concept shift. In [a], concept shift is defined as semantic changes, and in their experiments  CIFAR100 is split into 10 sub-datasets with increasing conceptual differences from CIFAR10 classes. For example, pickup truck (CIFAR100) is much closer to truck (CIFAR10) than sunflowers (CIFAR100) semantically. (2) Our example in Figure 1 is similar to [b]. In [b], they gave an example using the Waterbird dataset, where the backgrounds (land and water) are spuriously correlated with class labels (waterbirds and landbirds). Notice that when using images for fairness learning, since inputs are images followed by class and sensitive labels, some aspects of the image features may inadvertently encapsulate information related to the sensitive label.
>
> [W2] The fairness level in Figure 2 is calculated using demographic parity (DP) using the evaluation metric used in experiments. The equation of DP is proposed in Appendix C.3.
>
> [W3] In this paper, we use a binary sensitive attribute and classes just as a simple case for idea illustration, as many fairness works did, such as [b]. This can be easily extended to multinary cases by either changing the group fairness notion or taking the maximum difference of DP over each class/subgroup.
>
> [W4, W5] The AUC (defined in Appendix C.3) in this work refers to a statistically consistent measure of predictive strength by comparing outcomes between sensitive subgroups. Notice that this is different from the AUC used for classification and this metric is commonly used for fairness learning, see [c,d,e].
>
> [W6] Dependence between $Y$ and $Z$ is to reveal the bias with respect to the dataset unfairness. Dependence between $\hat{Y}$ and $Z$ refers to algorithmic fairness with respect to the model prediction. Dataset unfairness is used as a reference to compare with the algorithmic fairness for performance improvement.
>
> [W7] Such works will be discussed in an updated version.
>
> [Q1] In Problem 1, without the constraint, it is equivalent to a domain generalization (DG) problem stated in some DG works, such as [f,g]. A fair constraint can be perceived as the pursuit of optimal model parameters within a fair parameter space, which is derived from the real domain after removing sensitive attributes.
>
> [Q2] DP as a commonly used group fairness metric refers to the likelihood of a positive outcome being the same across different groups. We use DP only because this metric is well-known in fairness machine learning and is considered in many related works. Some other fair metrics, such as equalized odds, can also be applied.
>
> [Q3] The reduction of AUC indicates the sum ranks of predicted outcomes between sensitive subgroups are similar. This AUC metric (defined in Appendix C.3) is not the same as the one commonly used in binary classification based on TPR and FPR. The intuition behind this AUC is based on the nonparametric test, Mann–Whitney U test, in which for randomly selected values $\mathbf{x}_1$ and $\mathbf{x}_2$  from two sensitive subgroups, a fair condition is defined as the probability of $\mathbf{x}_1$ being greater than $\mathbf{x}_2$ is equal to the probability of $\mathbf{x}_2$ being greater than $\mathbf{x}_1$, i.e. AUC=0.5.
>
>
>
> [a] Tian et al., Exploring Covariate and Concept Shift for Out-of-Distribution Detection, NeurIPS 2021
>
> [b] Creager et al., Environment Inference for Invariant Learning, ICML 2021
>
> [c] Calders et al., Controlling Attribute Effect in Linear Regression, ICDM 2013
>
> [d] Zhao et al., Rank-Based Multi-task Learning For Fair Regression, ICDM 2020
>
> [e] Ling et al., AUC: a statistically consistent and more discriminating measure than accuracy. IJCAI 2003
>
> [f] Zhang et al., Towards principled disentanglement for domain generalization. CVPR 2022
>
> [g] Robey et al.,  Model-based domain generalization. NeurIPS 2021

---

### Meta-Review · Area_Chair_MjjH · 2023-12-06

**Metareview:**

This paper studies domain generalization and fairness simultaneously under covariate and sensitive dependence shifts. The sensitive dependence shift means there is a spurious relation between the sensitive attribute and the label. The authors formulate the problem as a minimax optimization problem with demographic parity constraint. Experiments were done on several benchmark datasets.

Strength: the reviewers all agree that this problem is essential to the fairness community.

Weakness: the major concerns after rebuttal are 1) writing: for example, several reviewers mention the usage of sensitive attributes and spurious correlations interchangeably, which makes the paper less rigorous; 2) technical soundness: for example, there exists a gap between theory and the proposed method.

**Justification For Why Not Higher Score:**

This paper is below the acceptance line. Both the scores and the discussion reflects that.

**Justification For Why Not Lower Score:**

N/A

---

### Decision · Program_Chairs · 2024-01-16

Reject